# How Do Environmental Regulations and Outward Foreign Direct Investment Impact the Green Total Factor Productivity in China? A Mediating Effect Test Based on Provincial Panel Data

**DOI:** 10.3390/ijerph192315717

**Published:** 2022-11-25

**Authors:** Decai Tang, Zhangming Shan, Junxia He, Ziqian Zhao

**Affiliations:** School of Management Science and Engineering, Nanjing University of Information Science & Technology, Nanjing 210044, China

**Keywords:** environmental regulation, green total factor productivity, outward foreign direct investment, mediating effect test, regional heterogeneity

## Abstract

This paper investigates the impact of two types of environmental regulations (ERs), command-and-control environmental regulation (CACER) and market-incentive environmental regulation (MIER), on green total factor productivity (GTFP) through outward foreign direct investment (OFDI) in 30 provinces in China for the period of 2006–2019. The Global Malmquist–Luenberger (GML) Index based on non-radial directional distance function (NDDF) considering undesired outputs is used to measure GTFP growth at the provincial level. To explore the mediating effect of OFDI, the two-step econometric model and the non-linear mediating effect model are employed. The empirical results show that CACER has an inverted U-shaped impact on OFDI and a U-shaped impact on GTFP, while MIER has a linearly positive effect on OFDI and GTFP. The current intensity of CACER lies on the left side of the inflection point of the U-shaped curve. OFDI significantly positively influences the increase in GTFP and is a significant mediating variable in the relationship between ERs and GTFP. Moreover, the introduction of OFDI delays the appearance of the inflection point. Further analysis, taking into account the regional heterogeneity, indicates that the inverted U-shaped and U-shaped curve is still valid in the eastern and western area and that the mediating effect of OFDI on ERs in the western area is stronger than that in the eastern area. Based on these conclusions, policy implications are provided to improve GTFP in China.

## 1. Introduction

Since the reform and opening-up in 1978, China has achieved 9.8% annual growth in gross domestic product (GDP), with per capita GDP growing at an annual rate of 8.5% [1]. However, such an economic miracle comes at the price of severe environmental pollution due to large amounts of energy consumption and pollution emissions [2,3]. According to the 2018 Global Environmental Performance Index Report, China’s environmental performance index ranked 120th, far behind its economic position in the world, which makes it imperative for the country to accelerate its industrial green transformation [4]. The green transformation process can only be completed by increasing green productivity [5]. To evaluate the quality of regional green development, green total factor productivity (GTFP) has been widely employed by scholars in the field [6,7].

Energy saving and pollution reduction have become the strategic theme of China’s Thirteenth Five-Year Plan [8], in which the Chinese government declared a series of goals such as reducing the energy consumption per unit GDP and the levels of pollutant emissions by 15% and 10% to 15% [9], respectively. To achieve these goals, China has implemented environmental regulations and other policies to reduce the environmental pollution caused by industrial production. After continuous reform and improvement, an environmental regulatory system compromising both command-and-control and market-incentive environmental regulations has been formed. Command-and-control environmental regulation (CACER) and market-incentive environmental regulation (MIER) are also two commonly mentioned regulation tools in the academic field, of which CACER refers to mandatory environmental policies such as laws, regulations and institutions drafted by legislative or administrative departments [4], while MIER is led by the market: the government can motivate enterprises to reduce pollution emissions via the building emissions trading market or by charging environment taxes and pollution discharge fees [10]; hence, it is another way to take advantage of an economic incentive mechanism to guide the market. China’s environmental regulation policy is improving [11]. Table 1 summarizes the major environmental regulation policies implemented in China. Efforts made by the Chinese government clearly reflect its determination to strengthen the environmental governance of the whole country. Then, what influences do ERs have on the GTFP of China’s industrial sector? How do different types of environmental regulation tools affect GTFP? Is there any mediating mechanism in the influencing process? These questions remain to be answered with careful investigation.

Moreover, the Chinese government has adopted the going-out strategy to encourage overseas investment to accelerate the domestic green transformation and industrial upgrading [12]. In fact, outward foreign direct investment (OFDI) from developing and emerging markets accounts for a considerable share of the entire world, which, according to the 2019 World Investment Report, amounted to USD 418B in 2018, some 45% of the world’s total value. China, as the largest economy among the developing countries, is no exception. The 2020 Statistical Bulletin of China’s Outward Investment launched by the Chinese Ministry of Commerce shows that, in 2020, the net flow of China’s outward foreign direct investment (OFDI) reached USD 153.71B, an increase of 12.3% compared with 2019. By the end of 2020, 28 thousand Chinese domestic investors had established 45 thousand overseas enterprises in 189 countries (regions) around the world, and the accumulated OFDI net stock reached USD 2.58T. The World Investment Report 2021 by UNCTAD showed that global FDI outflows reached USD 0.74T in 2020, and the year-end stock was USD 39.35T, which means that China’s OFDI flows and stock in 2020 accounted for 20.2% and 6.6% of the global total, respectively, ranking first among all countries (regions) in terms of OFDI flows and third in terms of stock. This trend has attracted research interest to investigate the characteristics of OFDI from developing markets and its impact on the local economy [13]. OFDI is becoming an increasingly important part of promoting China’s green development [14]. Liu et al. argue that OFDI has a positive environmental influence on domestic green innovation by transferring advanced green technology back to the home country, which is called the reverse green technology spillover effect. Can OFDI really improve the quality of domestic economic development? There are large amounts of research discussing the environmental effects of international investments, known as “pollution halo” and “pollution haven” [15,16,17,18,19,20,21,22]. However, these environmental effects have mostly been discussed from the perspective of the host country, and few studies pay attention to the impact of OFDI on the home country’s environmental quality. Therefore, questions arise as to whether the environmental regulations from the home country affect its OFDI and, further, GTFP. Moreover, is there any difference between the impacts that different types of environmental regulation instruments have on China’s OFDI, thus, further affecting the GTFP? How do these interactions differ by regional heterogeneity? Making clear these problems can not only provide references for the Chinese government to implement reasonable environmental regulation policies to promote green transformation, but also offer insights into how to properly use OFDI to achieve sustainable development.

This paper aims to provide a better understanding of the relationships between two different types of ERs (CACER and MIER), OFDI and the GTFP. First, we measure the GTFP by the Global Malmquist–Luenberger (GML) index and its decomposition by employing the non-radial directional distance function with undesirable outputs. Second, we illustrate the impact of ERs on GTFP through OFDI within a unified theoretical framework by establishing the two-step econometric model and the non-linear mediating effect model. We also testify to the robustness of the econometric model by introducing both a difference-generalized method of moments (Diff-GMM) and system-generalized method of moments (Sys-GMM) models. Third, we examine the impacts that both CACER and MIER have on the GTFP and test the mediating effect by introducing OFDI into the model. We also explore the non-linear effect of ERs on both OFDI and GTFP and, respectively, quantify the mediating effect of OFDI on the relationship between ERs and GTFP. Finally, the economic zone is divided into the eastern, central, and western parts of China to study the influence that comes with the location heterogeneity on the relationship between ERs and GTFP through OFDI, aiming to provide a theoretical basis to formulate differentiated environmental regulations and overseas investment policies.

Compared with previous research, the marginal contributions of this paper are: First, we measure the provincial GTFP by employing the non-radial directional distance function with undesirable outputs and calculate the GML index and its decomposition. Second, instead of employing a single measurement to study the relationship between the stringency of the environmental regulation and GTFP, we explore the impact of different types of ERs, CACER, and MIER, exerting on GTFP, such that the implementation effect of ERs can be accurately differentiated and more specified implications can be derived. Third, we test the mediating effect of OFDI by introducing the two-step regression model and the non-linear mediating model. By making analytical derivations and coefficients estimations, we also calculate the quantified mediating effect and its proportion in the total effect. The rest of this paper is organized as follows: Section 2 reviews related research. Section 3 provides a theoretical framework and hypotheses. Section 4 conducts research design, including the measurement of GTFP and the econometric models. Section 5 presents empirical results and discussions. Section 6 concludes and provides the relevant policy recommendations.

## 2. Literature Review

### 2.1. The Measurement of Green Total Factor Productivity

The concept of green total factor productivity is derived from total factor productivity (TFP), which reflects the quality of economic growth by taking into account the impacts of both technological progress and efficiency improvement. In the early TFP literature, scholars took labor and capital as inputs and desirable output, such as GDP, to measure the TFP of countries and regions [5,23,24,25,26,27]. However, since the neglect of resource saving and emissions reduction could result in an overestimation of TFP, scholars began to incorporate energy inputs and environmental outputs into the measurement framework. Meanwhile, as environmental issues and public concerns have become more prominent, the GTFP index has become an important indicator to measure the green growth of the economy. GTFP can be measured in two ways: one is parametric, using the Stochastic Frontier Analysis (SFA) method, and the other is the non-parametric method, Data Envelopment Analysis (DEA), based on linear programming. Using panel data from the 1995–2010 period, Zhang and Ye [28] measured the GTFP growth of 29 provinces in China by the SFA method. As for the DEA model, both radial and non-radial directional distance functions (DDF) are employed by scholars to estimate the GTFP. For example, Wu et al. [29] calculate the GTFP of 46 “Belt and Road” countries spanning from 2003 to 2016 by applying the DDF-based ML index, while Cheng et al. [30] study the total energy efficiency and its spatial convergence in Chinese provinces by employing the non-radial DDF meta-frontier method. Compared with the SFA method, the non-parametric DEA method is more popular for the capacity to deal with multiple inputs and outputs without requiring the monetary values for environmental variables [31]. Therefore, this paper adopts the latter. Different from the above literature, this paper measures the provincial GTFP of China by a global MML index calculation based on the non-radial DDF with undesirable outputs (please see details in Section 4.1).

### 2.2. Environmental Regulation and Its Impacts on GTFP and OFDI

Since environmental pollution has become one of the most important public concerns, scholars have conducted various studies on environmental regulation and its impacts on GTFP, and conclusions differ from one between studies. First, there are two streams of research on whether environmental regulation benefits or hurts green total factor productivity. One stream believes that implementing ERs can inhibit TFP [11,32]. Tang et al. [11] believe that implementing ERs internalizes the negative externalities of enterprises and forces them to turn to production that is not optimal for them, thus, hurting their innovation capability and market competitiveness. Cai and Ye (2020) [32] evaluate the impact of China’s new environment protection law through a Difference-In-Difference (DID) model and find that the law produced a 3.94% decline in the TFP of the studied enterprises, thus, verifying the 2020 study by Tang et al. Conversely, the other research stream claims that environmental regulation would encourage enterprises to improve their efficiency by investing in innovation activities [33,34,35,36], which is known as the “Porter Hypothesis” in the field. Zhang et al. [37] claim that the more stringent the environmental regulation intensity, the higher the TFP growth rate in China. Mi et al. [2] extended the Porter Hypothesis and investigated the impacts of other dimensions of regulations besides environmental regulation on Chinese economic development. Wang et al. [21] employed an environmental policy stringency index to measure environmental regulation and an extended SBM-DDF model to measure green productivity growth and found that a strong version of the Porter Hypothesis is validated in promoting the green growth of OECD countries. Peng et al. [38] observe that the productivity-enhancing effect occurs with environmental regulations. By separating R&D into environmentally induced R&D and production R&D, Zhang [36] highlights that environmentally induced R&D is the driver of GTFP. Moreover, the non-linear relationship between environmental regulation and GTFP is also proposed and emphasized in some research. Li et al. [39] find an inverted U-shaped relationship between environmental regulation and GTFP. Wang et al. [21] confirm the same conclusion with the dynamic GMM model. Wu and Lin [40] found that in the context of China’s iron and steel industry, the “Porter Hypothesis” is verified, and the U-shaped relationship exists between environmental regulation and efficiency. Guo and Yuan [41] study the heterogeneous influence of different types of environmental regulation tools, i.e., command-and-control regulation and market-based regulation and confirm the Porter effect of environmental policies on energy efficiency. They also find that market-based regulation is more effective for the current reality in China. Since few scholars pay attention to the heterogeneous functions of different types of environmental regulation and their impacts on GTFP, this paper aims to fill this gap.

In terms of the influence of ERs on OFDI, Zhou et al. [42] identify a threshold effect of ERs and suggest that only when ERs achieve a specific level can OFDI improve the GTFP for the home country. Yu et al. [43] explore the relationship between the emissions trading system (market-based environmental regulation) and carbon leakage through OFDI in developing countries and show that the emissions trading system accelerated both the depth and breadth of OFDI from China. Pan et al. [44] discuss the role that ERs play in influencing OFDI and its promotion of the home country’s carbon intensity. The same topic is also investigated by Hao et al. [45] in exploring environmental quality. By employing both static and dynamic panel data models, Liu et al. [46] provide statistical evidence that there is a moderating effect of ERs on the reverse green technology spillover of OFDI, and further empirical results show that ERs produce a significant crowding-out effect on such spillover effect in eastern regions of China.

Based on the previous research, we can see that literature on the impact of different types of environmental regulations on GTFP is not very common, and studies exploring the impact of ERs on OFDI are also rare. Few studies investigate the relationship between ERs and GTFP through OFDI. Therefore, this paper aims to explore both the linear and non-linear relationships between ERs and GTFP through OFDI. We also quantify the mediating effect of OFDI on the impact of ERs on GTFP, and none of the above literature employs the same quantitative methods as our study.

### 2.3. OFDI and Its Impact on Green Economy Development

Traditional economic research focuses on the promoting effect of foreign direct investment (FDI) on the host countries’ technological development and social welfare [42,47]. Some literature also investigates the reversal technological spillover effect of OFDI on home countries [48], of which technological development and innovation are identified as the key contributors to economic growth [49,50].

Regarding OFDI and its impact, the related research is mainly conducted from both the micro and macro perspectives. At the micro firm level, Li et al. [51] hold that OFDI has become one of the pivotal factors that seek technological sources for Chinese enterprises. Nugent and Lu [52] explore the investing motivation of Chinese firms under the Belt and Road Initiative. Chen et al. [53] document the mechanism of different OFDI patterns affecting Chinese firms’ internal resource allocation efficiency. Zhang and Kong [54] examine the impacts and mechanisms of both traditional and new energy policies on the OFDI of A-share listed companies in China and find strong promotion effects. Valacchi et al. [55] investigate the impact of OFDI on the patent activity of Greenfield multinational enterprises and find that both capital investment and jobs created through Greenfield FDI boost the innovation activity of multinational companies, and such an effect is more significant in industries with higher patent intensities. By applying the Propensity-Score-Method-Difference-In-Differences (PSM-DID) method, Dong et al. [13] explore the reversal technology spillovers OFDI on Chinese firms and propose three potential transmission channels for OFDI to improve Chinese firms’ innovation abilities. At the macro country level, Liu et al. [15] introduce panel threshold regression models to investigate the OFDI–green innovation relationship of China from the home country’s perspective and decompose the relationship into scale, composition, and technique effects. Li et al., Zhang et al. and Tang et al. [56,57,58] theoretically analyze and empirically test the impact of OFDI on the home country’s global value chain upgrading. Luo et al. [59] claim that OFDI might lead to massive investment and a long-term investment cycle, given the total financial constraints; this may, in turn, decrease the R&D investment of a certain company, thus, hindering the green innovation activities of the enterprise. Pan et al. [44] developed a panel smooth transition regression model and studied the non-linear effects of OFDI on China’s total factor energy efficiency. They employ industrial structure, opening-up level, and human capital as transition variables and find that the thresholds for the above variables are 37.59%, 36.11%, and 8.33%, respectively.

However, scholars are divided on whether OFDI can promote productivity. By employing data from Germany, Herzer [60] claims that OFDI has a positive relationship with output and productivity. Zhao et al. [61] also believe that the technical spillover effect exists for Chinese enterprises and benefits their productivity improvement. Driffield et al. [62] study the impact of OFDI on British productivity and prove that investments in R&D and labor-intensive areas help improve productivity. In general, however, the positive effect is not significant. Contrarily, Bitzer and Kerekes [63] conclude that IFDI (inward FDI) has a significant productivity-promoting effect, while OFDI does not; their conclusion comes from the empirical industrial-level data of 17 OECD countries. There are also scholars investigating the reasons for drawing different conclusions. Li et al. [64] investigate the impact of OFDI on Chinese domestic innovation and the mediating effects of absorptive capacity, foreign investment, and market competition. Based on the industrial-level data of the Chinese manufacturing sector, Huang and Zhang [65] find that the influence of OFDI differs due to differences in ownership nature, absorptive ability, and international strategy adjustment.

Although there has been extensive literature on OFDI and its impact on green economy development, few studies have discussed the impact of OFDI on the environment from a home country perspective, nor do they investigate whether the heterogeneous environmental regulations impact the growth of GTFP through regulating the reverse spillover of OFDI. To bridge these research gaps, this paper builds a two-step econometric model to test the mediating effects of OFDI in the process of ERs influencing GTFP.

## 3. Theoretical Framework and Research Hypotheses

The theoretical mechanism of the relationship between ERs, OFDI, and GTFP is constructed to empirically test the effects of ERs on GTFP through OFDI. Figure 1 depicts the impacting mechanism.

### 3.1. Environmental Regulations in China

Compared to developed countries, China is lagging in the introduction and implementation of environmental protection policies and currently has three types of tools: command and control, market-incentive, and informal environmental regulation.

#### 3.1.1. Command-and-Control Regulation

Command-and-control environmental regulation (CACER) is one of the most important tools for the government to target pollution emissions. It mainly conducts macro-control and guidance to the market and enterprises through the formulation of policies, the promulgation of decrees, administrative participation, and other means. Its characteristics are high compulsion, resulting in the subject of negative externality failing to comply with the standard regulations. The administration will restrict production, shut down and even order polluting industrial enterprises to quit in order to achieve “survival of the fittest”. Later, China enacted the “Air Pollution Prevention and Control Law”, the New “Environmental Protection Law”, and other laws to maintain the macro direction.

#### 3.1.2. Market-Based Regulation

The use of market-incentive environmental regulation (MIER) instruments in China differs significantly from that of developed countries. As for environmental tax, the Third Plenary Session of the 18th Central Committee of the Communist Party of China (CPC) in November 2013 deployed the “implementation of the principle of statutory taxation”, and in March 2015, clarified the tax type, tax rate, and levy management system, which has been in effect since 1 January 2018. As for tradable emission permits, since the beginning of the practice of emissions trading in 1987, the scope of pilot projects has been increasing yearly. In terms of subsidies for energy saving and emission reduction, government support is strong, but the overall application is insufficient. Given this, the emission charging system is the main market-based incentive tool in China.

#### 3.1.3. Informal Regulation

Informal environmental regulation is a tool that is usually not mandatory. The government gives moral advice to the regulated. The main feature of informal environmental regulation is environmental information and public participation. This has become the main means for the public to proactively safeguard their environmental interests through the NPC and CPPCC meetings, hearings, letters, visits and complaints, supervision by public opinion, and environmentally friendly social publicity.

### 3.2. The Theoretical Mechanism of ERs on OFDI and GTFP

#### 3.2.1. The Crowding-Out and Technology Seeking Effects of ERs on OFDI

On the one hand, as globalization enables the production process to be expanded to foreign countries to gain a country-specific advantage, ERs is becoming an increasingly important factor that influences enterprises’ investment decisions. That is, heavy polluting firms migrate from countries with stringent ERs to countries with relaxed ERs; we call this the crowding-out effect of ERs on OFDI. When the stringency of ERs in the home country is high, which means high pollutant emissions costs, enterprises within the home country will have enough motivation to invest more in foreign countries to avoid the punishment asserted by the local government. On the other hand, high ER standards mean that firms must adopt clean technology to improve their production process to meet the emission standards, due to which technological innovation is necessary. This is when advanced foreign technology can be transferred to the home country, which we call the technology seeking effect of ERs on OFDI. In fact, when pressured by ERs, investors tend to acquire advanced clean technology in the process of OFDI and take the technology as compensation for emission disadvantages in the home country [46]. In this context, we propose the following Hypothesis 1.

**Hypothesis** **1.***ERs have a positive effect on OFDI, and such an effect may vary by different types of ERs*.

#### 3.2.2. The Promoting and Preventing Effects of ERs on GTFP through OFDI

First, the implementation of ERs means extra costs for industrial enterprises, which is mainly reflected in either paying environmental taxes for environmental pollution or purchasing environmental protection equipment to reduce pollution emissions. To compensate for these extra costs, enterprises themselves would choose to expand the production scale to obtain the scale effect; GTFP as a whole would, therefore, be improved. In fact, OFDI is one of the pivotal sources of advanced technology—including clean technology—for reducing emissions, because it allows firms to learn by doing. Since the environmental policy is also an important market signal conveying the direction of industry development, ERs can also attract enterprises with long-term visions with green capital and technical talents to enter the market. This, in the long run, would also contribute to the improvement of GTFP. In other words, ERs can promote GTFP through OFDI. Meanwhile, the exact opposite situation could happen as well, i.e., ERs may also hurt GTFP through OFDI. As the stringent environmental policy requires industrial firms to increase environmental investments to meet ER standards, and this may divide part of the investment that could have been productive and profit-making, the improvement of productivity and GTFP will, thus, be hindered. Therefore, this paper assumes that ERs not only have a direct effect on GTFP, but may also increase it by encouraging OFDI. Under these circumstances, we propose the following Hypothesis 2.

**Hypothesis** **2.***ERs have both positive and negative effects on GTFP through OFDI, and the mediating effect of OFDI is significant*.

Moreover, since different regions endowing different resources and economic development could react differently to the same level of ER stringency, the OFDI seeking to transfer clean technology could also be spatially heterogeneous under the influence of ERs. In addition, since the mechanism of different types of environmental regulation is also different, the mediating effect varies by different types of ERs. Therefore, Hypothesis 3 is proposed.

**Hypothesis** **3.***The mediating effect of OFDI features different characteristics in terms of different types of ERs and region heterogeneity*.

### 3.3. The Theoretical Mechanism of OFDI on GTFP

#### The Green Spillover Effect of OFDI on GTFP

China’s going-out strategy is proposed to accelerate the green transformation of the economy. Multi-national companies (MNCs) from China, however, do not possess very strong intellectual property and technological resources. To overcome this disadvantage, they pursue advanced production factors—for example, technology and talents—via approaches such as opening up R&D centers and seeking corporations with institutions in host countries to improve environmental innovation and energy efficiency at a relatively lower cost. As a result, green spillovers derived from OFDI can be realized. In addition, expanding OFDI in foreign countries intensifies the competition with the horizontal industries from host countries, which in turn leads to observation and imitation of foreign technologies, i.e., knowledge spillover. This kind of spillover, in a similar vein, can also be transferred to other domestic firms through their parent firm undertaking OFDI. Beyond this kind of transfer, the same imitation happens among firms of the same sector in the home country. That is, advanced technologies and environmental experiences are integrated and assimilated by the whole sector. In brief, OFDI improves GTFP in both direct and indirect ways. Hence, we argue the following Hypothesis 4.

**Hypothesis** **4.***OFDI has a green spillover effect on GTFP, thus, impacting it positively*.

## 4. Research Design

### 4.1. The GTFP Measurement

To explore the proposed hypotheses, our research design starts with the GTFP measurement. The DEA method is widely employed by scholars to evaluate the input–output efficiency of decision-making units (DMUs). Efficiency is measured by the distance between the production unit and the frontier. Compared with the directional distance function (DDF), the non-directional distance function (NDDF) posits that outputs are not always produced proportionately by inputs, thus, it is more consistent with real practice. According to the differences between distance functions, there are radial efficiency and non-radial efficiency, and this paper employs the latter measurement. Chung et al. [66] developed the Malmquist–Luenberger (ML) productivity index to measure TFP by taking the undesirable output, pollution, into account. However, the traditional ML productivity index presumes that all production units face the same production frontier, thus, failing to reflect the differences in actual production technologies between DMUs. Therefore, based on the non-radial directional distance function, considering different resource endowments, economic structures, and the development of different regions [67], we employ the MML index. This index takes into account the heterogeneous factor of the group production frontier to measure the GTFP of 30 provinces in China. To overcome the infeasible problems encountered during the calculation, the global production technology set is constructed to involve all the environmental and technical elements. Finally, the Global MML index (GML index) is introduced to measure GTFP in this paper.

#### 4.1.1. Non-Radial Directional Distance Function (NDDF) with Undesirable Outputs

Assume *K* DMUs at a time *t*, and each DMU produces *m* types of desirable outputs 
$y=\left(y_{1},\;y_{2},\;\ldots ,\;y_{m}\right)\in R_{+}^{M}$ and J types of undesirable outputs $b=\left(b_{1},\;b_{2},\;\ldots ,\;b_{j}\right)\in R_{+}^{J}$ by using n types of inputs $x=\left(x_{1},\;x_{2},\;\ldots ,\;x_{n}\right)\in R_{+}^{N}$. The technology set is:(1)T={(x,y,b): x can produce (y,b)}

Hence, the production technology set can be described as follows:(2)$$P^{t}\left(x^{t}\right)=\{\left(x^{t},y^{t},b^{t}\right)|\sum_{k=1}^{K}\lambda_{k}^{t}x_{kn}^{t}\leq x_{kn}^{t},\forall n;\sum_{k=1}^{K}\lambda_{k}^{t}y_{km}^{t}\leq y_{km}^{t},\forall m;\sum_{k=1}^{K}\lambda_{k}^{t}b_{kj}^{t}\leq b_{kj}^{t},\forall j\}$$

Chung et al. (1997) [66] defined the DDF as:(3)$$D_{R}\left(x,y,b;g\right)=\sup\{\beta :\left(\left(x,y,b\right)+\beta g\right)\in T\}$$
where $g=\left(g_{x},g_{y},g_{b}\right)\in R_{-}^{J}\times R_{+}^{M}\times R_{-}^{N}$ is a pre-assigned nonzero vector specifying the direction from the data point, (x,y,b), to the production frontier. Since $D_{R}$ is based on the radial efficiency measurement, which ignores the fact that not all the inputs and desirable (undesirable) outputs are expanded (contracted) at the same rate, it may overestimate the efficiency once there are nonzero slacks [68]; a non-radial DDF is defined in this paper as:(4)$$D_{NR}\left(x,y,b,g\right)=\sup\{w^{T}\beta :\left(\left(x,y,b\right)+\mathrm{diag}\left(\beta \right)\cdot g\right)\in T\}$$
where w denotes a normalized weight vector relevant to the number of inputs and outputs and $\beta =\left(\beta_{x},\beta_{y},\beta_{b}\right)\in R^{J}\times R^{M}\times R^{N}$ denotes the vector of the scaling factors [69]. Then, the non-radial DDF measurement of inefficiency can be estimated by solving the following linear programming problem:(5)$$\begin{array}{l} D_{NR}\left(x,y,b;g\right)=\max w^{T}\beta \\ s.t.\;\left\{ \begin{array}{l} \sum_{k=1}^{K}\lambda_{k}x_{k}\leq x+\mathrm{diag}\left(\beta_{x}\right)\cdot g_{x}\\ \sum_{k=1}^{K}\lambda_{k}y_{k}\leq y+\mathrm{diag}\left(\beta_{y}\right)\cdot g_{y}\\ \sum_{k=1}^{K}\lambda_{k}b_{k}\leq b+\mathrm{diag}\left(\beta_{b}\right)\cdot g_{b}\\ \beta \geq 0\\ \sum_{k=1}^{K}\lambda_{k}=1;\;\lambda_{k}\geq 0,\;k=1,\ldots ,K \end{array}\right. \end{array}$$

Following Oh [70], $D_{NR}$ is based on the global technology set, including all DMUs and all time periods. $\lambda_{k}$ is the weight variable, $\sum_{k=1}^{K}\lambda_{k}=1\;$ means that the production possibility set is a variable return to scale (VRS); without the constraint, the production possibility set would be constant to scale (CRS).

#### 4.1.2. GML Index and Its Decomposition

Based on the NDDF with undesirable outputs, by employing the global production set, the GML index can be expressed as:(6)$$\begin{array}{l} GML=\sqrt{\frac{1+D_{NR}^{t+1}\left(x^{t+1},y^{t+1},b^{t+1};g\right)}{1+D_{NR}^{t+1}\left(x^{t},y^{t},b^{t};g\right)}}\\ \times \sqrt{\frac{1+D_{NR}^{t}\left(x^{t+1},y^{t+1},b^{t+1};g\right)}{1+D_{NR}^{t}\left(x^{t},y^{t},b^{t};g\right)}} \end{array}$$

It can be further decomposed into efficiency change (GMLEC), technology change (progress) (GMLTC), and scale returns change (GMLSC):(7)$$GMLEC=\frac{D_{NRV}^{t+1}\left(x^{t+1},y^{t+1},b^{t+1};g\right)}{D_{NRV}^{t}\left(x^{t},y^{t},b^{t};g\right)}$$
(8)$$GMLTC=\sqrt{\frac{D_{NRV}^{t}\left(x^{t+1},y^{t+1},b^{t+1};g\right)}{D_{NRV}^{t+1}\left(x^{t+1},y^{t+1},b^{t+1};g\right)}\times \frac{D_{NRV}^{t}\left(x^{t},y^{t},b^{t};g\right)}{D_{NRV}^{t+1}\left(x^{t},y^{t},b^{t};g\right)}}$$
(9)$$GMLSC=\sqrt{\frac{D_{NRC}^{t}\left(x^{t+1},y^{t+1},b^{t+1};g\right)/D_{NRV}^{t}\left(x^{t+1},y^{t+1},b^{t+1};g\right)}{D_{NRC}^{t}\left(x^{t},y^{t},b^{t};g\right)/D_{NRV}^{t}\left(x^{t},y^{t},b^{t};g\right)}\times \frac{D_{NRC}^{t+1}\left(x^{t+1},y^{t+1},b^{t+1};g\right)/D_{NRV}^{t+1}\left(x^{t+1},y^{t+1},b^{t+1};g\right)}{D_{NRC}^{t+1}\left(x^{t},y^{t},b^{t};g\right)/D_{NRV}^{t+1}\left(x^{t},y^{t},b^{t};g\right)}}$$

The details about the decomposition, I refer to Banker et al. [71].

### 4.2. Model, Variables, and Data Sources

#### 4.2.1. The Econometric Model

To detect the transmission mechanism of ERs on GTFP through OFDI, we first set the following two-step econometric model:(10)$$GTFP_{it}=\alpha_{0}+\alpha_{1}CACER_{it}+\alpha_{2}MIER_{it}+\sum_{j=1}^{k}\beta_{j}X_{it}^{j}+f_{i}+f_{t}+\varepsilon_{it}$$
where i and t indicate the province and the year; $f_{i}$ and $f_{t}$ refer to the individual fixed effect and time fixed effect; $\varepsilon_{it}$ represents the random disturbance term; $GTFP_{it}$ denotes the GTFP of each province; $CACER_{it}$ and $MIER_{it}$ are two environmental regulation indicators representing command-and-control and market-incentive ERs, respectively. $X_{it}$ stand for the other control variables, including research and development (R&D), industrial structure (ISTR), energy structure (ESTR), foreign direct investment (FDI), human capital level (HU), α identifies the impacts of ERs on GTFP, η identifies the impacts of ERs on OFID, ξ represents ERs contribute to GTFP through OFDI, $\varepsilon_{it}^{\prime }$ and $\varepsilon_{it}^{\prime \prime }$ represent the OFID and the impacts of ERs and OFDI on random disturbance term of GTFPAs the relationship between ERs and GTFP may be non-linear, the quadratic term of CACER and MIER is added to the model (10) in the following way:(11)$$GTFP_{it}=\alpha_{0}+\alpha_{1}CACER_{it}+\alpha_{2}MIER_{it}+\alpha_{3}CACER_{{}_{it}}^{2}+\alpha_{4}MIER_{it}^{2}+\sum_{j=1}^{k}\beta_{j}X_{it}^{j}+f_{i}+f_{t}+\varepsilon_{it}$$

Since ERs not only have a direct effect on GTFP but may also contribute to GTFP through OFDI, we set the following model:(12)$$OFDI_{it}=\eta_{0}+\eta_{1}CACER_{it}+\eta_{2}MIER_{it}+\eta_{3}CACER_{it}^{2}+\eta_{4}MIER_{it}^{2}+\sum_{j=1}^{k}\beta_{j}X_{it}^{j}+f_{i}^{\prime }+f_{t}^{\prime }+\varepsilon_{it}^{\prime }$$
(13)$$GTFP_{it}=\xi_{0}+\xi_{1}CACER_{it}+\xi_{2}MIER_{it}+\xi_{3}CACER_{it}^{2}+\xi_{4}MIER_{it}^{2}+\xi_{5}OFDI_{it}+\sum_{j=1}^{k}\beta_{j}X_{it}^{j}+f_{i}^{\prime \prime }+f_{t}^{\prime \prime }+\varepsilon_{it}^{\prime \prime }$$
where $OFDI_{it}$ indicates the OFDI of each province. Model (12) also considers the non-linear effect of ERs on OFDI. The mediating effect model is depicted in the following Figure 2.

If both ERs and OFDI have a significant impact on GTFP and ERs alone have a significant impact on OFDI, then it can be concluded that OFDI is a significant mediating variable for the relationship between ERs and GTFP. Moreover, if the impact of ERs on GTFP is significant without OFDI while insignificant when adding the variable, then OFDI has a complete mediating effect; otherwise, the mediating effect is partial [72]. Please refer to Appendix A for the detailed procedures of the mediating effect test.

#### 4.2.2. The Non-Linear Mediating Effect Model

The non-linear mediating effect model is employed to calculate the effect of ERs on GTFP through OFDI. We first define the total effects of ERs on GTFP as TEFF by calculating the partial derivatives of GTFP on CACER and MIER and obtain $TEFF_{C}=\frac{\partial GTFP}{\partial CACER}$; $TEFF_{M}=\frac{\partial GTFP}{\partial MIER}$. According to model (11), we have:(14)$$TEFF_{C}=\frac{\partial GTFP}{\partial CACER}=\alpha_{1}+2\alpha_{3}CACER$$
(15)$$TEFF_{M}=\frac{\partial GTFP}{\partial MIER}=\alpha_{2}+2\alpha_{4}MIER$$

In a similar way, we define the instantaneous indirect effect of ERs on GTFP through OFDI as MEFF. Then, MEFF can be calculated by multiplying the partial derivatives of OFDI to ERs with that of GTFP to ERs as follows:(16)$$MEFF_{C}=\frac{\partial OFDI}{\partial CACER}\cdot \frac{\partial GTFP}{\partial OFDI}=\eta_{1}\xi_{5}+2\eta_{3}\xi_{5}CACER$$
(17)$$MEFF_{M}=\frac{\partial OFDI}{\partial MIER}\cdot \frac{\partial GTFP}{\partial OFDI}=\eta_{2}\xi_{5}+2\eta_{4}\xi_{5}MIER$$
where $\frac{\partial OFDI}{\partial CACER}$ and $\frac{\partial OFDI}{\partial MIER}$ are drawn from the model (12) and $\frac{\partial GTFP}{\partial OFDI}$ from the model (13). Finally, the proportion of the mediating effects to the total effects can be calculated in the following way:(18)$$\gamma_{C}=\frac{\eta_{1}\xi_{5}+2\eta_{3}\xi_{5}CACER}{\alpha_{1}+2\alpha_{3}CACER}$$
(19)$$\gamma_{M}=\frac{\eta_{2}\xi_{5}+2\eta_{4}\xi_{5}MIER}{\alpha_{2}+2\alpha_{4}MIER}$$

The derivation of the model is the same way as Yang et al. [73].

#### 4.2.3. Variables and Data Processing

##### Dependent Variable

The NDDF model and GML index are employed to measure the GTFP of China’s 30 provinces. Note that the measured GML index is the growth rate of GTFP instead of GTFP itself; following Qiu et al. [74], we convert the index into the actual value of the GTFP. All variables, including the input and output variables, are displayed in Table 2. (1) Labor: Considering the availability of the statistical data, this paper employs the annual average number of industrial employees as the labor input. (2) Capital: the total fixed assets investments are adopted to represent the capital input. (3) Energy: industrial energy consumption is taken as the proxy variable of the energy input. (4) Desirable output: we choose the real gross domestic production (GDP) of each province to express the desirable output. (5) Undesirable output: industrial waste water emissions, industrial waste gas emissions, and industrial solid waste emissions are selected as undesirable outputs.

##### Core Independent Variable

(1) Command-and-control environmental regulation (CACER): In China, CACER mainly includes “greenhouse gas emission control” since the Thirteenth Five-Year Plan and “regulations on the administration of environmental protection” [59]. However, this indicator cannot objectively reflect the policy outcomes of environmental regulations. The term “pollution treatment investment” refers to the funding used to create fixed assets in the process of industrial pollution treatment, including investments in “three simultaneous” environmental protection projects, new and old industrial pollution source control projects, and the construction of urban environmental infrastructure. The term “three simultaneous” refers to a project in which the main component of the project and the infrastructures for preventing pollution are concurrently developed, built, and employed. Most environmental protection investment is derived from government expenditure [74]. Since it can result in rapid improvements in environmental performance, CACER is, at present, the most widely used environmental regulation tool in most countries in the world. Many studies have used environmental protection investment to represent CACER [5,59,76,77]. Following their lead, this paper employs the same proxy.

(2) Market-incentive environmental regulation (MIER): MIER is mainly guided by the market. The Chinese government imposes pollutant discharge fees to achieve pollution reduction goals through the market mechanism. Pollutant discharge fees refer to a governmental charge per unit of emissions by measuring the number of emissions of a company; this is a compliance market mechanism. The government uses these fees to regulate the way in which it can correct the illegal behavior of enterprises, who try to avoid, and abuse the responsibility of pollution control. It gives enterprises the right to choose freely, and, through the collection of sewage charges to guide the reduction in environmental pollution behavior of environmental regulatory instruments, is in accordance with the “polluter pays” principle, which is designed to have a restraining effect on the environmental pollution behavior of enterprises, while affording them the right to make their own production and management decisions, and ultimately, achieve the purpose of protecting the environment, saving energy, and reducing emissions. Therefore, in this paper, MIER is measured by the total amount of pollutant discharge fees levied on each province [4,59,77,78].

(3) Outward foreign direct investment (OFDI): The ratio of each province’s OFDI stock (converted into RMB by the exchange rate of the current year) to its GDP is calculated to represent the level of outward foreign direct investment. The reason that we employ the stock rather than the flow of OFDI is that the latter is dynamic and can be affected by various uncertain factors [59].

##### Control Variables

(1) Research and development (R&D): Since research and development promotes technical progress, which helps enterprises pursue pollution control and emissions reduction, it is an important and powerful engine for GTFP growth. The proportion of internal R&D expenditure to each province’s total GDP is employed to represent the R&D investment level of a region. (2) Industrial structure (ISTR): In China, more than 90% of CO_2_ emissions are from industrial energy consumption, of which the second industry accounts for a large proportion [79,80]. Moreover, industries featuring less energy consumption and pollution emissions but a greater increase in value can improve technical efficiency, thus, improving GTFP. Therefore, we use the ratio of the added value in the tertiary industry to the added value in the second industry to quantify ISTR. (3) Energy consumption structure (ESTR): Since environmental pollution is mainly from coal consumption and hinders green development, we take the ratio of coal consumption to total energy consumption as the measurement of ESTR. (4) Foreign direct investment (FDI): FDI has mixed effects on green productivity. On the one hand, it produces a technology spillover effect, helping improve GTFP by introducing green technology; on the other hand, it leads to serious environmental pollution by transferring heavy pollution industry to host countries. In this paper, we express FDI by the proportion of each province’s FDI to its GDP. (5) Human capital level (HU): Human capital level reflects the education level of employees in a region. Well-educated personnel with rich knowledge and skills provide absorptive capacity for technology, and thus, technological progress. As such, the proportion of college students in the total urban employment population of each province is employed to indicate HU. A description of the independent variables is shown in Table 3.

#### 4.2.4. Data Sources

To ensure both the validity and availability of the research information, we set the study period as 2006 to 2019. Due to the lack of key indicators, Taiwan, Hong Kong, Macao, and Tibet are excluded, and we collected the data from 30 provinces of China. Finally, a strongly balanced data set is obtained. With regard to the data to calculate the GTFP of each province, labor and capital input data are collected from the China Statistical Year Book; energy input data are collected from the China Energy Statistical Year Book; desired output data are collected from the China Statistical Year Book; and undesired outputs data are collected from the China Environmental Statistical Year Book. Based on all these data, we calculated the provincial GTFP growth by the Stata programming proposed by Wang et al. [69], as detailed in Section 4.1. Moreover, environmental regulations data are collected from the China Statistical Year Book and China Environmental Statistical Year Book; outward foreign direct investment data are collected from the Statistical Bulletin of China’s Outward Foreign Direct Investment; research and development data are collected from the China Statistical Year Book on Science and Technology; industrial structure, foreign direct investment, and human capital level data are collected from the China Statistical Year Book; and energy consumption structure data are collected from the China Energy Statistical Year Book. The descriptive statistics are shown in Table 4. To alleviate the influence of heteroscedasticity and multicollinearity, variables with the original absolute quantity are logarithmically processed.

Table 5 reports the Variance Inflation Factor (VIF) of the independent variables and the correlation test between variables. As can be seen in Table 5, the largest VIF value among all variables is 1.92, much lower than 10. Thus, we conclude that there is no multicollinearity problem in the model.

## 5. Results and Discussions

### 5.1. Results and Analysis of GTFP and Its Decomposition

Table 6 shows the average annual growth rate of provincial GTFP during the period 2006–2019. We can see that the average GTFPs of the three regions and the whole country are all lower than 1, meaning that GTFP in all regions show a decreasing trend. This is because, in the past, China achieved rapid economic growth at the expense of an efficient use of resources and environmental protection. There is now a shift in the way the economy develops. With the pursuit of high-quality development, ERs increase the cost of enterprises. The western region is decreasing the most and the eastern region ranks second, closely followed by the central region. Specifically, GTFP development varies from province to province. Hainan boasts the highest GTFP growth rate among all the provinces (15.09%), followed by Shanghai (7.27%), Henan (4.89%), and Chongqing (2.87%). Contrarily, Xinjiang has the lowest GTFP growth rate (−15.86%). According to Qiu et al. [74], this is because the economic development in Xinjiang features heavy resource consumption, thus, resulting in high polluting emissions. Regarding the decomposition indexes, the average value of GMLTC and GMLSC all exceed 1 at both the national and regional levels, while the average value of GMLEC in most provinces is lower than 1.0000. Beijing and Jiangsu are the only regions whose GMLEC equals 1, indicating that Beijing and Jiangsu are the only places that achieve the highest technical efficiency and realize the optimal inputs–outputs allocation. Since the average values of GMLTC are all bigger than that of GMLSC, the conclusion can be made that technological progress contributes more to GTFP than scale returns. The GMLTC index peaks at 18.57% in Shanghai, followed by Chongqing (14.55%), Henan (11.73%), and Zhejiang (10.59%). This conclusion is also consistent with Feng et al. [81], who claim that technological progress is the main driving force of GTFP growth. Meanwhile, only Liaoning, Shandong, Guangdong, Guangxi, Sichuan, and Shaanxi are lower than 1 in GMLTC (technical efficiency). As for the levels of technological progress (GMLEC) in different regions, central China ranks first and western China ranks last, suggesting that technological progress is associated with economic development and a series of joint factors such as resources endowment and development patterns. This is because, in terms of the level of economic development alone, provinces in the eastern region are superior to provinces in the central region.

### 5.2. Estimated Results of the Two-Step Econometric Model

In this section, the two-step econometric model is employed to empirically verify the proposed hypotheses. To make the estimations credible, we performed both the pool regression and the two-way fixed effect regression (fixed effect thereafter to simplify the expression) models simultaneously. Considering the potential non-linear effects of the independent variables, the quadratic terms of CACER and MIER are, respectively, introduced to the models. In addition, to overcome the endogeneity caused by the bilateral causal between ERs and GTFP, both the Difference-GMM and System-GMM methods are employed, with the first order lag term of the dependent variables, OFDI and GTFP, as an instrumental variable (IV), respectively. Based on the regional heterogeneity, analysis in terms of eastern, central, and western provinces is provided as well. Finally, the mediating effects are quantified in detail by the estimated coefficients.

#### 5.2.1. The Effect of ERs on OFDI

To estimate the effect of ERs on GTFP, model (12) is first tested, and the result is given in Table 7. Columns (1)–(3) report the pool regression results, and columns (4)–(6) perform the fixed effect regression. Moreover, both the linear and non-linear relationship between ERs and OFDI is taken into account and added to the model. As shown in Table 7, in terms of the linear relationship between CACER and OFDI, the estimation results indicate that every unit increase in CACER causes a 0.471–0.636 unit increase for OFDI, with a 1% significant level (see columns (2) and (5)), meaning that CACER can effectively improve the level of OFDI. In the same vein, the positive impact of MIER on OFDI ranges from 0.002 to 0.003 units but with a lower significance level (see columns (3) and (6)). As such, we can conclude that the impact of CACER is greater than that of MIER.

Regarding the non-linear relationship between ERs and OFDI, when respectively introducing the quadratic terms of CACER and MIER into the model, we can find that the primary coefficient of CACER is still significantly positive, while its quadratic coefficient is significantly negative (see column (3)), indicating that CACER does not have a simple, linear relationship with OFDI, but rather an inverted U-shaped one. In other words, when the CACER intensity is under a certain level, enhancing CACER can help promote OFDI. Once the CACER intensity exceeds the inflection point, however, it will hinder OFDI. This conclusion is also consistent with Wang et al. [21] and Yang et al. [73]. Referring to Yang et al. [73], based on the estimation results, by setting $-0.050x^{2}+1.781x-17.2=0.636x-10.726$ (where x represents the CACER intensity and y represents the OFDI level in the pool regression model), we approximately obtain the inflection point, as shown in the last row of Table 7. The inflection point of CACER in the fixed effect model is the same. When the intersection cannot be found, i.e., there is no real solution to the equation, we take the inflection point of the estimated quadratic function. According to the calculated inflection point, when CACER is lower than 12.7238, it can promote OFDI; once the CACER intensity exceeds the value, it will hinder the promotion of OFDI. The conclusion indicates that only well-designed environmental regulation policies can promote OFDI in an appropriate way. Following a similar lead, MIER does not exhibit the same feature because all the quadratic coefficients of the MIER approach are 0. Based on these observations, theoretical Hypothesis 1 is verified.

With respect to the estimated results of the control variables, industrial structure (ISTR) has a significantly positive influence on OFDI, while research and development (R&D), energy structure (ESTR), and human capital level (HU) have a significantly negative impact on OFDI, suggesting that an advanced industrial structure can accelerate the outflow of capital, while the investment in R&D slows down the progress. Moreover, the coefficients of energy structure indicate that the higher the proportion of carbon consumption in the total energy consumption, the lowers the outflow of capital. This indirectly verifies the theoretical assumption in Section 3.1 that when the stringency of ERs in the home country is low, industrial enterprises will lack the motivation of capital outflow and consume the home country’s energy at a lower cost, which in turn leads to high carbon consumption. Finally, human capital labor can also reduce OFDI. This is because the enhancement of the human capital level can substitute OFDI to some extent by stimulating innovation and advanced technologies, as we propose in Section 3.1.

#### 5.2.2. The Effects of ERs and OFDI on GTFP

The estimated results of the impacts of ERs and OFDI on GTFP are reported in Table 8. Columns (1), (2), (5), and (6) show the estimated results without introducing OFDI. Similar to Table 7, we also compare the situation with and without the quadratic terms of ERs and find that, in this context, CACER also has a significant non-linear relationship with GTFP, while MIER does not. As the primary term of CACER is negative while its quadratic term is positive (see columns (1) and (5)), the non-linear relationship features a U-shaped curve. This means that CACER has a negative impact on GTFP when it is at a lower level, and when it exceeds a certain level, this impact could be positive. The impact of MIER on OFDI is significantly positive. Since the quadratic terms of MIER all approach 0, the positive impact can be taken as a monotonous linear increase (see columns (1), (5), and (6)). Columns (3), (4), (7), and (8) are the estimated results when OFDI is introduced. Columns (4) and (8) assume that both ERs and OFDI have a linear effect on GTFP, while columns (3) and (7) assume that ERs and OFDI have, respectively, non-linear and linear effects on GTFP. The results show that when introducing OFDI, the non-linear effect of CACER on GTFP is still valid. The impact of CACER on GTFP after OFDI is introduced. GTFP is significantly positive, indicating that, to some extent, OFDI can improve GTFP. This conclusion is consistent with Zhou et al. [42], who claim that OFDI only introduces green benefits to the home country under certain circumstances. Therefore, Hypothesis 4 is verified. The last row reports the inflection points of CACER, with columns (1) and (5) reporting when OFDI is not introduced, and columns (3) and (7) reporting when OFDI is introduced. As can be seen from the calculated results, the inflection point of the U-shaped curve is in the interval between 10.1053 and 12.9464. Since the average intensity of CACER in China is 11.85 (see Table 4), right within the interval, we conclude that the CACER level at this stage needs to be further improved to accelerate the promotion of GTFP.

In order to show the inflection points more intuitively, Figure 3 depicts the impact of CACER on GTFP before and after OFDI is introduced. As can be observed from Figure 3, the inflection point of the U-shaped curve is delayed after OFDI is introduced. The reason lies in the impact of CACER on GTFP through OFDI, which suggests that with the out flow of OFDI, the promotion of GTFP in the home country will need more investment in command-and-control environmental regulation. So far, the first half of Hypothesis 2 and Hypothesis 3 are verified. Moreover, the estimation results of the control variables show that energy structure has a significant positive impact on GTFP, while foreign direct investment and human capital level have a negative impact on GTFP, although less significantly. Finally, the impact of industrial structure on GTFP is totally insignificant, indicating that it is not the main reason for the change of GTFP.

### 5.3. Endogenous and GMM Estimation

ERs are usually regarded as exogenous variables with a single relationship with OFDI and GTFP. However, OFDI may effectively reduce pollution emissions, thus, reducing the requirements for ERs intensity [73,82]. Moreover, when GTFP continues to increase, the pollution emissions level returns to a reasonable range, and no additional ERs need to be intensified. Therefore, potential bi-directional casual effects may exist between ERs and GTFP. To overcome the endogenous problem and avoid the estimation bias caused by the traditional regression methods, this paper employs the first-order lag terms of the dependent variables (L.OFDI and L.GTFP in Table 9) as instrumental variables (IV), and both the Sys-GMM and Diff-GMM methods are utilized to test the estimated results in Table 7 and Table 8. As shown in Table 9, the first order lag term of OFDI is significantly positive, while that of GTFP is significantly negative, suggesting that both the previous OFDI and GTFP have a continuous impact on current OFDI and GTFP and that OFDI growth is a continuous dynamic accumulation, while GTFP growth exhibits dynamic reduction. Compared with Table 7 and Table 8, the signs and significance of the impacts of ERs on OFDI and GTFP remain the same, thus, ensuring the robustness of the previous estimations.

Finally, the Hansen test for over-identification indicates that all IVs are valid (p>0.1). Meanwhile, the Arellano–Bond test verifies that the first-order autocorrelation is significantly true (p<0.1) and the second-order autocorrelation is significantly not true (p>0.1), indicating that both Sys-GMM and Diff-GMM can overcome the endogenous problem and further ensure the robustness of the previous conclusions.

### 5.4. Heterogeneity Analysis

Due to differences in the resources endowment and economic development level, different regions pay different degrees of attention to the local environment. This would result in significant spatial heterogeneity regarding the investment policy and environmental policy [74]. In other words, ERs and OFDI may exert different impacts on GTFP within different regions, and the mediating effect of OFDI may as well be heterogeneous across regions. To make comparisons, this paper divides all of the selected provinces into three regions by their geographical locations. Figure 4 shows the kernel density curves of CACER, MIER, OFDI, and GTFP in the three regions, from which we can see that the kernel density curves of CACER and OFDI in the eastern area exhibit a clear right-tailed distribution, thus, showing a higher level in both CACER and OFDI than in western and central provinces, while the kernel density curve of MIER shows a different picture. With respect to GTFP, the three regions are all unimodal and have a right distribution, with the convergence interval largest in the central region and narrowest in the western region. In light of the analysis of the kernel density curves, we have a good reason to divide the studied provinces into three regions and conduct a further comparative analysis.

#### 5.4.1. The Effect of ERs on OFDI across Regions

The estimation results of the impact of ERs on OFDI across regions are shown in Table A3 and Table A4. For consistency with the previous analysis, both pool regression and individual fixed effect regression are performed. The linear estimates show that CACER has a significantly positive effect on OFDI in the eastern and western regions, while its impact in the central region is not significant (see columns (2), (5), (8)). Meanwhile, MIER has a significantly positive impact on OFDI in the western area (see columns (3), (6), (9)). According to the estimated results, every unit increase in CACER creates 0.173–0.746 units of increase in OFDI in the eastern area, while 0.674–0.962 units in the western area, suggesting that CACER has a greater impact on the latter area. Regarding the non-linear relationships between ERs and OFDI, the estimated results show that, in the eastern area, CACER has a significantly inverted U-shaped relationship with OFDI because the estimated coefficients in the western area are insignificant. (See columns (1) and (7)). In addition, in the centra area, CACER and OFDI approximately present a U-shaped relationship, but this is not significant (see column (4)). Following Section 5.2, we also calculated the inflection points of the non-linear relationship between CACER and OFDI in the eastern area and report them in the last row of Table A3 and Table A4. Hence, a more specific conclusion is made that, different from the average national level (11.9825–12.7238) when CACER in the eastern area is within the interval of 9.3635–17.17672, it can make a contribution to the promotion of OFDI. In other words, too low or too high a level of CACER will seriously hinder the promotion of OFDI in the eastern provinces.

In a similar vein, Table A3 and Table A4 also report the estimated results of the control variables, among which industrial structure has a significantly positive impact on OFDI across regions, while foreign direct investment and human capital level have a significantly negative correlation with OFDI in the central area, suggesting the regional heterogeneity of the impacts from the control variables.

#### 5.4.2. The Effect of ERs and OFDI on GTFP across Regions

The estimated results of the impact of ERs and OFDI on GTFP across regions are shown in Table A5 and Table A6. (1) When OFDI is not introduced, as can be seen from columns (2), (6), and (10), the linear relationship between CACER and GTFP is no longer significant, while MIER has a significantly positive effect on GTFP in the eastern area, indicating that MIER in the eastern area can effectively promote GTFP. When adding the quadratic term of CACER and MIER into the regression models, the observations remain the same that there exists no non-linear relationship between MIER and GTFP across three regions because the estimated coefficient of the quadratic terms of MIER is either insignificant or 0. Moreover, the U-shaped curve between CACER and GTFP can still be found at the regional level, and the relationship is significant in the eastern and western areas (see columns (1), (5), and (9)). This indicates that there are inflection points for the impact of CACER on GTFP in eastern and western China, and once CACER exceeds the inflection, it will significantly promote GTFP in these areas. To be more specific, we respectively calculated the value of the inflection point of the U-shaped curve by the aforementioned method. The results can be seen in Table A5 and Table A6, and the calculation details are shown in Appendix B. From Table A5 and Table A6, we can see that the negative impact of CACER on GTFP will become positive when it exceeds 13.5895 in the eastern area and 10.8506 in the western area. (2) When OFDI is introduced, the linear effects of CACER on GTFP are insignificant across regions, and the linear effect of MIER on GTFP is significantly positive in the western area (see columns (4), (8), and (12)). When the quadratic term of CACER is introduced into the regression models, the results show that the U-shaped relationship between CACER and GTFP in both the eastern and western areas is still significant (see columns (3) and (11)). Similarly, we reach the conclusion that the direction of the impact of CACER on GTFP will change when it exceeds 13.5895 in the eastern area and 11.1640 in the western area. This also suggests that, compared with the western area, GTFP in the eastern area is more influenced by CACER and OFDI. A more intuitive description can be found in Figure 5 and Figure 6. As shown in Figure 5 and Figure 6, after introducing OFDI, the inflection point of the U-shaped curve is delayed in the western area while approximately remaining the same. This indicates that more investment in CACER will be needed in the western area if OFDI is employed to accelerate the increase in GTFP. Based on the discussions, the second half of Hypothesis 3 is verified.

#### 5.4.3. Analysis of the Mediating Effect across Regions

The results presented in Table A3, Table A4, Table A5 and Table A6 indicate that OFDI is a partial mediating variable of the impact of CACER on GTFP in both eastern and western areas, and there is no mediating effect of OFDI in the central area. This is because before and after OFDI is introduced, the estimated coefficients of CACER are always significant in the eastern and western areas while insignificant in the central area (please refer to Appendix A for the mediating test procedure). Based on these observations, following Section 5.5, we also respectively calculated the total effect of ERs on GTFP ($TEFF_{C}$ and $TEFF_{M}$), the mediating effects of OFDI on CACER ($MEFF_{C}$) and MIER ($MEFF_{M}$), and the proportion of the mediating effect on the total effect (γ) in the eastern, central, and western areas. The results are shown in Table 10, from which we can see that the mediating effect of OFDI on CACER in the eastern and central area is in the interval of (−0.0086, 0.0003) and (−0.0066, 0.0238), respectively, suggesting that every unit increase in CACER can promote GTFP by 0–0.0086 units by promoting (hindering) OFDI in the eastern area. The central area follows in the same way. In contrast, only by promoting OFDI can every unit increase in CACER in the western area promote GTFP by 0.0066–0.0381 units.

In addition, OFDI is also a mediating variable of the impact of MIER on GTFP in the western area. The calculated results show that only by promoting OFDI can every unit increase in MIER promote GTFP by 0.0001–0.0004 units. This means that, taken as a whole, the mediating effect of MIER is much slighter than that of CACER. Moreover, the mediating effect of CACER in the eastern area accounts for 22.29–23.93% of the total effect, changing in a short spectrum, while the proportion of the mediating effect of CACER in the total effect in the western area ranges in the interval of 6.11–39.23%, which is much larger than the eastern area. This means that the mediating effect of CACER in the western area is stronger than in the eastern area. Regarding the mediating effect of MIER in the western area, the proportion is 30.33%. Since the coefficients of MIER2 are all 0 (see Table A1), there is no longer any interval here. For the integrity of the analysis, Table 10 lists all the calculated values. The results within the 10% significance level are shown in square brackets, and the results beyond the significance level are shown in parentheses.

### 5.5. Analysis of the Mediating Effect

According to columns (4)–(6) of Table 7 and columns (2)–(4) of Table 8, we can see that the mediating effect of OFDI is significant in the process of CACER influencing GTFP, while it is not significant in that of MIER influencing GTFP (the detailed mediating effect test procedure can be seen in Appendix A). Hence, the second half of Hypothesis 2 is verified. Nevertheless, the previous analysis does not specifically provide the quantified value of the impact of ERs on GTFP through OFDI, nor does it provide the proportion of the mediating effect to the total effect, as we defined in models (14)–(19). To this end, the mediating effects of OFDI on CACER ($MEFF_{C}$) and MIER ($MEFF_{M}$) are respectively calculated based on the defined models, as well as the total effect of ERs on GTFP ($TEFF_{C}$ and $TEFF_{M}$) and the proportion of the mediating effect in the total effect ($\gamma_{C}$ and $\gamma_{M}$). As shown in the first column of Table 10, the mediating effect of OFDI on CACER ranges from 0.0170 to 0.0256, meaning that every unit increase in CACER can increase GTFP growth by 0.0170 to 0.0256 units by stimulating OFDI. In parallel, the mediating effect accounts for 2.13–7.41% of the total effect, a ratio that sits in the interval of 1.39–22.68%. In a similar vein, the mediating effect of OFDI on MIER ranges from 0.0002 to 0.0003, much lower than CACER. When it comes to the proportion of the mediating effect in the total effect, MIER accounts for 22.68–2564.10%, a much larger interval than CACER, suggesting that there may be great regional heterogeneity in the mediating effect of MIER across provinces. In addition, the reason for the floating range of $\gamma_{M}$ and $\gamma_{M}\left(\bar{M}\pm \sigma_{M}\right)$ being the same lies in the coefficients of the quadratic terms of MIER all being 0 (as shown in Table 8). More details can be seen in Appendix B.

## 6. Conclusions and Policy Implications

Based on the panel data for China’s 30 provinces from 2006 to 2019, taking into account energy consumption and undesired outputs (pollution emissions), this paper begins by employing the NDDF model and GML index to calculate industrial GTFP and its decompositions. Then, a theoretical framework of ERs, OFDI, and GTFP is established and research hypotheses are put forward. Next, we construct a two-step econometric model and a non-linear mediating effect model to verify the effects of ERs and OFDI on GTFP. To ensure the robustness of the model, both Sys-GMM and Diff-GMM methods are used. Finally, the 30 provinces are divided into three regions to explore the impact of regional heterogeneity on the impacts of ERs on GTFP through OFDI, and the mediating effect of OFDI and its proportion in the total effect are quantified at both the national and regional levels. The conclusions are as follows.

First, GTFP growth varies from province to province, and technological progress is the main driving force for GTFP growth. According to the results, the average annual growth rate of GTFP in the whole country is 0.9444, indicating an urgent need to find methods for GTFP promotion. Regarding the regional difference, GTFP growth presents a pattern of progressive decrease from central China to eastern China and then to the western region. Second, at the national level, the results show that CACER has a significant inverted U-shaped impact on OFDI and GTFP, respectively. When CACER is lower than 10.1762, it can promote OFDI; otherwise, when it exceeds 12.7238, the increase will hinder OFDI. The situation for GTFP is in the same vein, and we find that the introduction of OFDI delays the inflection point. Different from CACER, MIER has a significantly linear increasing relationship with OFDI and GTFP. At the regional level, conclusions remain the same in the eastern and western areas. Third, OFDI has a significant mediating effect on the impacts of ERs on GTFP. We find that every unit increase in CACER can promote GTFP by 0.0170–0.0256 units by stimulating OFDI, and this value for MIER is 0.0002–0.0003. Moreover, the mediating effect of CACER and MIER accounts for 2.13–7.41% and 12.9–2564.10%, respectively. This indicates that MIER has a wider range of influence on GTFP and better policies than CACER, which is in line with the actual situation in China, so it needs to be well designed. Finally, the mediating effect of OFDI on both CACER and MIER in the western area is significant, as well as that of OFDI on CACER in the eastern area. Comparisons between different regions show that the mediating effect of CACER in the western area is stronger than in the eastern area, and only in the western area can MIER promote GTFP through OFDI, thus, providing another method for the improvement of GTFP in western provinces.

Based on these findings, policy implications are derived. First, the Chinese government should pay close attention to accelerating GTFP growth, especially the local government of the western area, with the most effective way being to increase investment in technological progress. Strengthening environmental governance and encouraging industrial enterprises to turn to more efficient and clean production is an urgent need. Second, a prudent environmental regulation policy is a key option for GTFP promotion in the coming period, and thus, it should be well-designed. On the one hand, the intensity of the command-and-control environmental regulation should be flexibly adjusted to cross the inflection point, in order to accelerate the promotion of GTFP; on the other hand, local governments should also pay more attention to the market-incentive environmental regulation. Moreover, local governments of each region should establish systematic environmental policies and make full use of the characteristics of different types of environmental regulation instruments based on their economic development and environmental situations. Third, the development of OFDI is also an effective way to improve GTFP by adjusting environmental regulations. Therefore, it is necessary for the Chinese government to make full use of its technology seeking and green spillover functions. Finally, considering the regional heterogeneity, eastern China should invest more in command-and-control environmental regulation, while western China should invest more in market-incentive environmental regulation. Moreover, combined with different types of environmental regulations, the level of OFDI in different regions should also be prudently considered based on the different conditions of each region.

This study also has the following limitations. First, due to data limitations, this paper does not investigate the differences between different types of OFDI. For example, different investment destination countries and industrial sectors could have different impacts on the GTFP of the home country, and hence, should not be generalized. In the future, we will consider collecting relevant data from a more specialized database to conduct a more detailed analysis. Second, this study explores the impact of ERs on GTFP through OFDI at the provincial level. However, the heterogeneity of industrial sectors also matters in this impacting progress. In fact, considering the differences in pollution emissions of various industrial sectors could draw more targeted conclusions and be more meaningful for the green development of specific industries. Third, although the econometric models in this paper take into account the non-linear effect of ERs, they ignore the spatial spillover effect of ERs between neighborhood regions because it is much more common for one local government to imitate the environmental regulations of another local government. Therefore, in the future, a spatial econometric model can be considered to verify such an influence.

## Figures and Tables

**Figure 1 ijerph-19-15717-f001:**
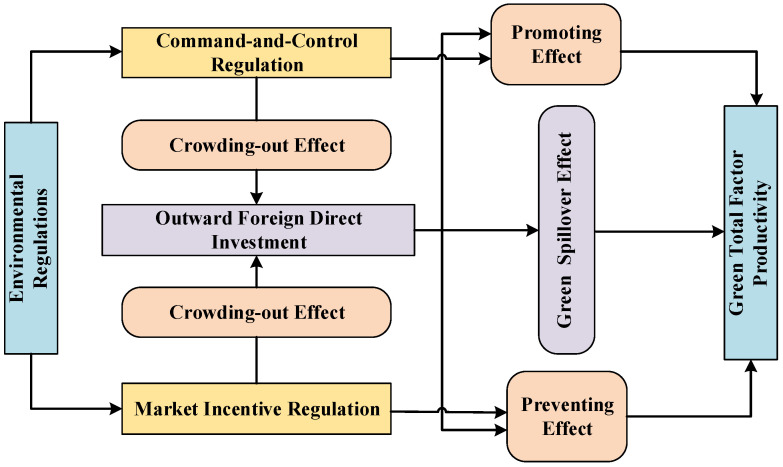
The impacting mechanism of ERs on GTFP through OFDI.

**Figure 2 ijerph-19-15717-f002:**
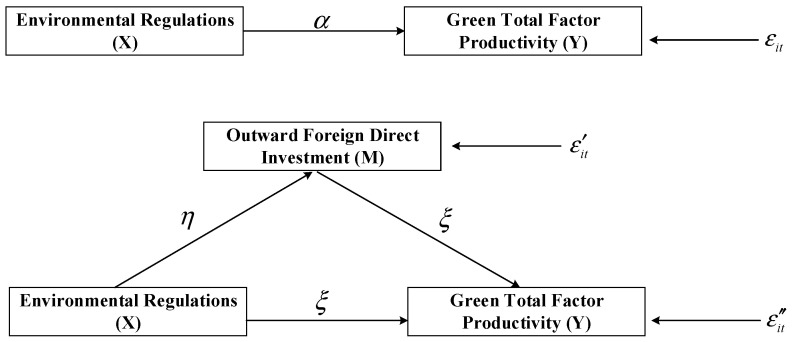
The mediating mechanism between ERs and GTFP through OFDI.

**Figure 3 ijerph-19-15717-f003:**
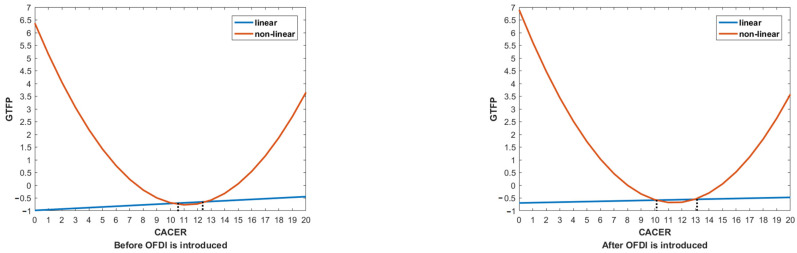
The inflection points for the impact of CACER on GTFP at the national level.

**Figure 4 ijerph-19-15717-f004:**
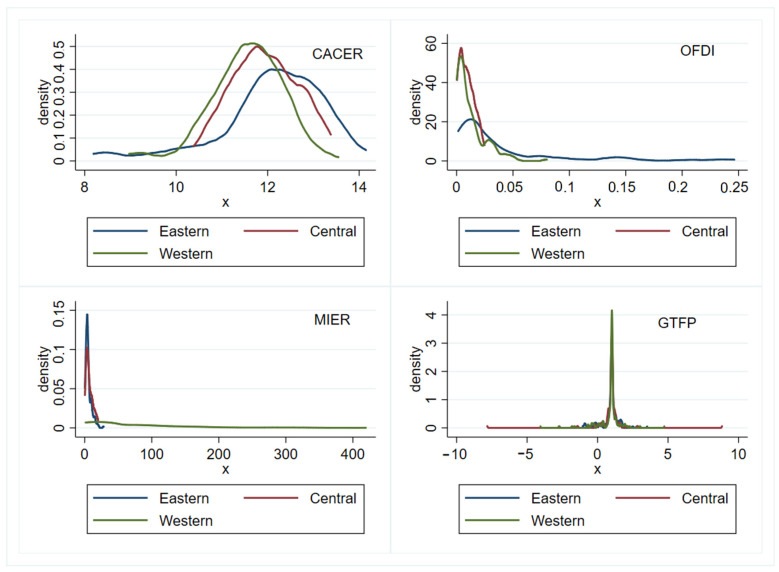
The kernel density curves of CACER, MIER, OFDI, and GTFP in different regions.

**Figure 5 ijerph-19-15717-f005:**
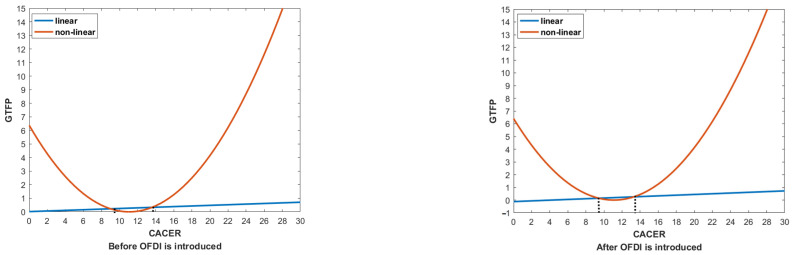
The inflection points for the impact of CACER on GTFP in eastern China.

**Figure 6 ijerph-19-15717-f006:**
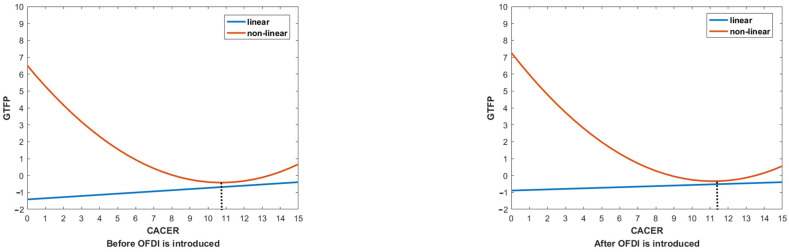
The inflection points for the impact of CACER on GTFP in western China.

**Table 1 ijerph-19-15717-t001:** The major environmental regulation policies in China.

Category	Strategy	Specific Policy
Command-and-control regulation	Permits	Pollution permit system (2016)
	Restricted use	Measures for the Administration of the Restricted Use of the Hazardous (2016)
	Emission standards	Pharmaceutical industry air emission standards (2019)
	Quotas	A quota system for electricity consumption from renewable sources (2019)
Market-incentive regulation	Pollution discharge fees	Collection Pay for Pollution Regulation (1982)
	Environmental tax	Environmental Protection Tax Law (2016)
	Producer Responsibility	Extended Producer Responsibility System Implementation Plan (2017)
	Emission trading	Carbon Emissions Trading Management Approach (2021)

Source: It has been souted out by the authors according to relevant materials.

**Table 2 ijerph-19-15717-t002:** GTFP input and output indicators.

Category	Variable	Measurement	Unit	Reference
Input Indicators	Labor	Annual average number of industrial employees	10,000 people	Qiu et al. [74]
	Capital	Total fixed assets investments	CNY 100M	Lee and Lee [75]
	Energy	Total industrial energy consumption	10^4^ tons	Cheng and Kong [4]
Desirable output	Desirable output	Real gross domestic product	CNY 100M	Qiu et al. [74]
	Undesirable output	Industrial waste water emissions	10^4^ tons	Lee and Lee [75]
		Industrial waste gas emissions	10^4^ tons	
		Industrial solid waste emissions	10^4^ tons	

**Table 3 ijerph-19-15717-t003:** Description of independent variables.

Variable	Definition
R&D	A country’s scientific and technological strength
ISTR	Secondary industry output to total GDP
ESTR	The ratio of coal consumption to total energy consumption
FDI	Foreign direct investment
HU	Reflects the education level of employees in a region

**Table 4 ijerph-19-15717-t004:** Descriptive statistics of core variables.

Variable	Obs	P50	Mean	Sta. Dev.	Min	Max
GTFP	420	1.010	0.922	0.880	−7.857	8.870
CACER	420	11.91	11.85	0.976	8.178	14.16
MIER	420	5.620	29.21	59.41	0.180	420.5
OFDI	420	0.0108	0.0217	0.0363	0.0001	0.247
R&D	420	0.0137	0.0166	0.0119	0.0022	0.0908
ISTR	420	0.891	1.067	0.609	0.500	5.169
ESTR	420	0.876	0.944	0.408	0.0248	2.461
FDI	420	0.141	0.163	0.110	0.0263	1.087
HU	420	0.215	0.404	0.511	0.0473	5.705

**Table 5 ijerph-19-15717-t005:** The correlation and multicollinearity test.

	VIF	GTFP	CACER	MIER	OFDI	R&D	ISTR	ESTR	FDI	HU
GTFP		1								
CACER	1.46	−0.04	1							
MIER	1.19	0.05	−0.15	1						
OFDI	2.11	0.06	−0.02	−0.04	1					
R&D	1.82	0.05	0.06	−0.18	0.38	1				
ISTR	2.50	0.09	−0.27	−0.04	0.65	0.53	1			
ESTR	1.63	−0.09	0.33	0.15	−0.37	−0.38	−0.42	1		
FDI	2.04	0.01	−0.14	−0.27	0.50	0.53	0.48	−0.49	1	
HU	1.21	−0.07	−0.01	−0.04	−0.24	−0.17	−0.13	0.27	−0.33	1

**Table 6 ijerph-19-15717-t006:** Average values of GML and its decomposition (2006–2019).

Region	Province	GML	GMLTC	GMLEC	GMLSC
Eastern China	Beijing	1.0000	1.0000	1.0000	1.0000
	Tianjin	0.8441	1.0000	0.9002	0.9439
	Hebei	0.9976	1.0670	0.9136	1.0170
	Liaoning	0.9119	0.9914	0.8171	1.0587
	Shanghai	1.0727	1.1857	0.8839	1.0031
	Jiangsu	0.8686	1.0000	1.0000	0.8686
	Zhejiang	0.9842	1.1059	0.8683	1.0100
	Fujian	0.9308	1.0741	0.8587	0.9981
	Shandong	0.8674	0.9943	0.8734	0.9997
	Guangdong	0.8421	0.9437	0.8883	1.0043
	Hainan	1.1509	1.0000	0.9566	1.1943
	Mean	0.9518	1.0329	0.9055	1.0089
Central China	Shanxi	0.9276	1.0000	0.9294	0.9982
	Jilin	0.9129	1.0865	0.8281	0.9983
	Heilongjiang	0.9192	1.0000	0.9308	0.9884
	Anhui	0.9697	1.0753	0.8946	0.9998
	Jiangxi	0.9512	1.0341	0.9158	1.0012
	Henan	1.0489	1.1173	0.9175	1.0141
	Hubei	0.9750	1.0828	0.9094	0.9828
	Hunan	0.9530	0.9701	0.8945	1.0885
	Mean	0.9572	1.0457	0.9025	1.0089
Western China	Inner Mongolia	0.9717	1.0774	0.8174	1.0769
	Guangxi	0.8991	0.9587	0.9254	1.0150
	Chongqing	1.0287	1.1455	0.8787	1.0045
	Sichuan	0.8910	0.9907	0.8773	1.0230
	Guizhou	0.9170	1.0521	0.8711	0.9937
	Yunnan	0.9562	1.0145	0.9377	1.0040
	Shaanxi	0.9557	0.9814	0.9665	1.0078
	Gansu	0.8676	1.0000	0.8879	0.9797
	Qinghai	0.8620	1.0000	0.8939	0.9681
	Ningxia	0.9758	1.0000	0.9488	1.0270
	Xinjiang	0.8414	1.0000	0.8539	0.9875
	Mean	0.9242	1.0200	0.8963	1.0079
	Full sample mean	0.9444	1.0329	0.9014	1.0086

**Table 7 ijerph-19-15717-t007:** Estimated results of the impact of ERs on OFDI.

	POOL			FE		
	(1)	(2)	(3)	(4)	(5)	(6)
CACER	1.781 ***	0.636 ***	1.748 **	1.431 *	0.471 ***	1.398 *
	(2.61)	(11.71)	(2.56)	(1.92)	(4.57)	(1.93)
MIER	0.006 ***	0.006 ***	0.003 ***	0.004	0.004	0.002
	(2.82)	(2.84)	(3.04)	(0.85)	(0.80)	(1.08)
CACER2	−0.050 *		−0.049 *	−0.042		−0.040
	(−1.68)		(−1.66)	(−1.34)		(−1.33)
MIER2	−0.000 *	−0.000 *		−0.000	−0.000	
	(−1.77)	(−1.75)		(−0.68)	(−0.63)	
R&D	−27.970 ***	−26.721 ***	−27.457 ***	−36.175 **	−36.608 **	−36.132 **
	(−5.44)	(−5.24)	(−5.34)	(−2.64)	(−2.66)	(−2.62)
ISTR	2.344 ***	2.325 ***	2.346 ***	2.956 ***	2.950 ***	2.948 ***
	(16.45)	(16.33)	(16.42)	(11.58)	(11.26)	(11.57)
ESTR	−0.532 ***	−0.552 ***	−0.516 ***	−0.942 *	−0.935 *	−0.978 *
	(−3.79)	(−3.93)	(−3.67)	(−1.77)	(−1.75)	(−1.89)
FDI	0.380 ***	0.351 ***	0.348 ***	−0.270	−0.296	−0.264
	(4.96)	(4.69)	(4.66)	(−1.04)	(−1.13)	(−0.98)
HU	−0.787 *	−0.684	−0.874 *	−3.814 ***	−3.800 ***	−3.870 ***
	(−1.74)	(−1.52)	(−1.94)	(−4.66)	(−4.68)	(−4.53)
Constant	−17.200 ***	−10.726 ***	−16.923 ***	−13.956 ***	−8.531 ***	−13.671 ***
	(−4.42)	(−17.61)	(−4.34)	(−3.26)	(−7.07)	(−3.32)
Observations	420	420	420	420	420	420
R-squared	0.560	0.557	0.556	0.642	0.640	0.640
Province	No	No	No	Yes	Yes	Yes
Year	No	No	No	Yes	Yes	Yes
Inflection point	10.1762	12.7238		10.2262	12.6309	

Robust t-statistics in parentheses; *** *p* < 0.01, ** *p* < 0.05, * *p* < 0.1.

**Table 8 ijerph-19-15717-t008:** Estimated results of the impact of ERs and OFDI on GTFP.

	POOL				FE			
	(1)	(2)	(3)	(4)	(5)	(6)	(7)	(8)
CACER	−0.587 **	0.007	−0.662 **	−0.018	−1.276 ***	0.027	−1.326 ***	0.011
	(−2.27)	(0.27)	(−2.35)	(−0.49)	(−3.77)	(0.69)	(−4.14)	(0.22)
MIER	0.002 *	0.002	0.002	0.002	0.004 **	0.004 **	0.003 **	0.004 **
	(1.72)	(1.67)	(1.42)	(1.40)	(2.31)	(2.61)	(2.16)	(2.44)
CACER2	0.026 **		0.028 **		0.057 ***		0.058 ***	
	(2.23)		(2.27)		(3.82)		(4.10)	
MIER2	−0.000	−0.000	−0.000	−0.000	−0.000 **	−0.000 **	−0.000 **	−0.000 **
	(−1.67)	(−1.66)	(−1.45)	(−1.46)	(−2.21)	(−2.47)	(−2.10)	(−2.36)
OFDI			0.043 **	0.039 **			0.039 ***	0.033 **
			(1.23)	(1.13)			(3.83)	(2.67)
RD	1.193	0.506	2.299	1.462	−0.713	−0.187	0.558	0.885
	(0.59)	(0.24)	(0.90)	(0.57)	(−0.12)	(−0.03)	(0.10)	(0.15)
ISTR	0.098	0.110	−0.002	0.021	0.228 *	0.232	0.114	0.138
	(1.24)	(1.38)	(−0.01)	(0.16)	(1.77)	(1.67)	(0.53)	(0.63)
ESTR	−0.123	−0.112	−0.100	−0.090	0.568 **	0.564 *	0.604 **	0.594 **
	(−1.60)	(−1.53)	(−1.17)	(−1.10)	(2.05)	(2.00)	(2.26)	(2.20)
FDI	−0.056	−0.040	−0.072 *	−0.054	−0.170 *	−0.134	−0.160 *	−0.124
	(−1.41)	(−1.01)	(−1.72)	(−1.24)	(−1.93)	(−1.33)	(−1.81)	(−1.25)
HU	−0.196	−0.247	−0.164	−0.222	−0.776 *	−0.788 *	−0.634	−0.669
	(−0.99)	(−1.20)	(−0.83)	(−1.08)	(−1.97)	(−1.95)	(−1.40)	(−1.44)
Constant	3.304 **	−0.060	4.039 **	0.363	6.373 ***	−0.985	6.894 ***	−0.694
	(2.35)	(−0.18)	(2.46)	(0.79)	(3.61)	(−1.58)	(4.12)	(−0.83)
Observations	420	420	420	420	420	420	420	420
R-squared	0.032	0.027	0.037	0.031	0.055	0.040	0.057	0.042
Province	No	No	No	No	Yes	Yes	Yes	Yes
Year	No	No	No	No	Yes	Yes	Yes	Yes
Inflection Point	10.3733 12.4729		11.8214		10.1806 12.6761		10.1053 12.9464	

Robust t-statistics in parentheses; *** *p* < 0.01, ** *p* < 0.05, * *p* < 0.1.

**Table 9 ijerph-19-15717-t009:** Robust test results.

	Sys-GMM	Diff-GMM	Sys-GMM	Diff-GMM
L.OFDI	0.664 ***	0.647 ***		
	(15.85)	(15.29)		
L.GTFP			−0.111 ***	−0.171 ***
			(−3.13)	(−4.67)
CACER	0.481 *	0.682 ***	−0.940 ***	−1.792 ***
	(1.77)	(2.64)	(−3.05)	(−4.11)
MIER	0.001	0.001	0.001	0.004 ***
	(1.37)	(0.86)	(1.24)	(2.68)
CACER2	−0.013	−0.023 **	0.040 ***	0.078 ***
	(−1.21)	(−2.19)	(2.99)	(4.07)
MIER2	−0.000	−0.000	−0.000	−0.000 ***
	(−0.88)	(−0.32)	(−1.37)	(−2.81)
OFDI			0.062 ***	0.064 ***
			(1.59)	(1.07)
RD	−6.313 *	1.805	2.241	4.524
	(−1.82)	(0.42)	(0.77)	(0.64)
ISTR	0.678 ***	0.927 ***	0.154	0.325
	(3.78)	(5.46)	(0.96)	(1.21)
ESTR	−0.199 **	−0.180	−0.042	0.732 **
	(−2.06)	(−0.73)	(−0.52)	(2.39)
FDI	0.084 *	−0.122	−0.121 **	−0.223 **
	(1.69)	(−1.35)	(−2.39)	(−2.03)
HU	−0.347	−0.853 ***	−0.330	−0.621
	(−1.36)	(−3.18)	(−1.41)	(−1.39)
Constant	−4.750 ***		5.575 ***	
	(−2.75)		(3.15)	
AR(1)	[0.000]	[0.000]	[0.028]	[0.061]
AR(2)	[0.663]	[0.636]	[0.484]	[0.404]
Hansen	[0.729]	[0.846]	[0.887]	[0.636]

Robust z-statistics in parentheses; *** *p* < 0.01, ** *p* < 0.05, * *p* < 0.1. *p* value in square brackets.

**Table 10 ijerph-19-15717-t010:** Mediating effects and total effects.

	Full Sample	Eastern Area	Central Area	Western Area
$TEFF_{C}$	[0.0749, 1.2032]	[−0.0010, 0.1053]	[−0.1348, 0.0023]	[0.097, 0.1077]
$TEFF_{M}$	[0.0020, 0.0040]	[−0.0038, −0.0028]	[−0.0063, 0.0035]	[0.003, 0.004]
$MEFF_{C}$	[0.0170, 0.0256]	[−0.0086, 0.0003]	(−0.0066, 0.0238)	[0.0066, 0.0381]
$MEFF_{M}$	[0.0002, 0.0003]	(0, 0.0004)	(−0.0022, 0.0001)	[0.0001, 0.0004]
$\gamma_{C}$	[2.13%, 7.41%]	[22.29%,23.93%]	(4.90%, 29.50%)	[6.11%, 39.23%]
$\gamma_{C}\left(\bar{C}\pm \sigma_{C}\right)$	[1.39%, 22.68%]	[−144.62%, 893.37%]	(−3.47%, 1026.51%)	[−39.19%, 4284.86%]
$\gamma_{M}$	[12.9%, 2564.10%]	(−4.03%, −0.55%)	(1.85%)	[30.33%]
$\gamma_{M}\left(\bar{M}\pm \sigma_{M}\right)$	[12.9%, 2564.10%]	(−13.45%, 9.89%)	(−29.50%, 35.43%)	[30.33%]

Note: $\bar{C}$ and $\bar{M}$ are the average values of CACER and MIER; $\sigma_{C}$ and $\sigma_{M}$ are the standard deviations of CACER and MIER, respectively. Since the coefficients in both the pool and fixed effect regression models are considered, quantified results here are all within certain intervals. Please see detailed calculations in Appendix B.

## Data Availability

Not applicable.

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
