# Peer review of "How Do Environmental Regulations and Outward Foreign Direct Investment Impact the Green Total Factor Productivity in China? A Mediating Effect Test Based on Provincial Panel Data"

_ijerph, 2022, doi:10.3390/ijerph192315717_

Round 1
Reviewer 1 Report
This paper reflects good, careful work and is well organized and clearly written. I will suggest some improvements in presentation -- in particular, addressing two issues that I believe are touched on but not adequately covered -- but I believe there is a major problem with the the execution of the project that needs to be addressed before smaller issues can be turned to.
The problem is with the proxies used to operationalize two of the three "core" independent variables: command-and-control environmental regulation (CACER) and market-incentive environmental regulation (MIER). It is a common problem when doing empirical research: the exact measure of the variable we are interested in is not available, so we choose an available proxy. Especially if the proxy is imperfect -- as is often the case -- it is important to keep in mind that if X is the proxy chosen for Y, the paper has not actually shown that Y determines Z; rather it has shown that X, which we argue is a proxy for Y, determines Z.
Here the problem is especially acute because the measurable variable chosen as a proxy for CACER, "environmental protection investment", is manifestly imperfect. One would expect that investment to increase in response to environmental regulations regardless of the exact form of the regulations. In particular, if we move from a system of no regulation to one of effluent taxes or permit trading, wouldn't we expect environmental protection investment to increase?
The distinction here -- and again, it is a major focus of the paper -- would seem to be at least in principal measurable by analyzing the specific environmental regulations imposed at the province level for the 30 provinces -- with perhaps intertemporal variation over the 14 year sample period. (I assume there is cross-section variation, since otherwise the research design seems unpromising.) Obviously this would also be imperfect, subject to judgment calls, difficult to measure distinctions among different levels of CACER and MIER, and so on, but at least it would be an attempt to focus more directly on the distinction of interest.
The measure of MIER, the total level of pollutant discharge fees levied, seems considerably better, and conceivably sufficient to make the distinction by itself: if a province collects no discharge fees, does it follow that whatever ER is used is not market-based? Maybe not, since setting up a marketable permit system might involve no "fees" to the government. But it is something to consider.
Two other broad concerns, both of which concern points that may be inferable from the presentations of the empirical results but which could use more direct discussion (if in fact they are inferable from the results): (1) Do the results tell us anything about whether MIER is superior to CACER in increasing GTFP?, and (2) Does Table 7 tell us anything about the significance and magnitude of including OFDI as a mediating factor for analyzing the impact of ER on GTFP?
Smaller points:
p. 4: The first paragraph of section 2.2 could use some work. I believe that on line 160, "GTFP" should be "TFP". More broadly, around line 175, my understanding is that the Porter hypothesis refers to TFP, innovation, and competitiveness only; I don't know of a "strong" version that refers to GTFP. If the authors do -- perhaps the cited Zhang, et al. paper? -- please explain and cite.
pp. 4-5: In section 2.3, is the ER under examination always in the home country only, or is the paper sometimes referring to ER in the host country? Again, just make clearer, please.
pp. 14-15: So from 2006-2019, GTFP decreased steadily in all regions? This sounds like a result that should receive more emphasis.
p. 15: Are provincial leaders rewarded, either formally or informally, for good environmental performance? Might this lead to biased reporting from the province to the center?
pp. 16-17: This is a pretty long paper. I would omit Figures 3-6.
p. 18, line 641: "columns (2)-(6)" should be "columns (4)-(6)".
p. 22: Where, exactly, are the results described in the first sentence shown? Which coefficient or coefficients should I be looking at?
p. 24, Figure 8: Please explain more clearly what these 4 curves mean, especially the bottom 2 with the single very high peaks.
p. 26, line 886: I thought the paper was examining OFDI as a mediating influence of ER on TFP. Where did this discussion of CACER as a mediating factor come from? But perhaps this is just an error of discussion.
p. 28, line 944: I would omit the statement that a policy implication of the paper is that the government should be providing tax incentives to encourage OFDI. If the rationale is to increase GTFP, surely this indirect instrument should be compared with more direct instruments such as tougher ER or incentives for PC research before reaching such a policy conclusion.
Author Response
Dear reviewer, thank you very much for your valuable comments and suggestions. According to your proposal, we have revised the paper, to get your approval.
Point 1: Here the problem is especially acute because the measurable variable chosen as a proxy for CACER, "environmental protection investment", is manifestly imperfect. One would expect that investment to increase in response to environmental regulations regardless of the exact form of the regulations. In particular, if we move from a system of no regulation to one of effluent taxes or permit trading, wouldn't we expect environmental protection investment to increase?
Response 1: In the absence of ER, the increase of environmental investment is conducive to green total factor productivity (GTFP), which is recognized by all of us. However, in the absence of ER, the direct benefit of environmental investment is small. However, in the case of ER, they will increase environmental protection investment when faced with the environmental performance of the superior government, so this "environmental protection investment" is reasonable to a certain extent. And we also refer to some of the following literature.
Hamamoto, 2006[ Hamamoto M. Environmental regulation and the productivity of Japanese manufacturing industries [J]. Resource and energy economics, 2006, 28(4): 299-312.];
Xie et al., 2017[ Xie R, Yuan Y, Huang J. Different types of environmental regulations and heterogeneous influence on “green” productivity: evidence from China [J]. Ecological Economics, 2017, 132: 104-112.]; Qiu et al., 2021; Luo et al., 2021),
Point 2: The distinction here -- and again, it is a major focus of the paper -- would seem to be at least in principal measurable by analyzing the specific environmental regulations imposed at the province level for the 30 provinces -- with perhaps intertemporal variation over the 14 year sample period. (I assume there is cross-section variation, since otherwise the research design seems unpromising.) Obviously this would also be imperfect, subject to judgment calls, difficult to measure distinctions among different levels of CACER and MIER, and so on, but at least it would be an attempt to focus more directly on the distinction of interest.
Response 2:Thank you for your suggestions. We also think their differences are very interesting, but We want to take them as my new research focus in the future. We believe We can write a very good article by comparing their differences.
Point 3:The measure of MIER, the total level of pollutant discharge fees levied, seems considerably better, and conceivably sufficient to make the distinction by itself: if a province collects no discharge fees, does it follow that whatever ER is used is not market-based? Maybe not, since setting up a marketable permit system might involve no "fees" to the government. But it is something to consider
Response 3:Setting up a marketable permit system might involve no "fees" to the government you raised is a very interesting topic, and We will pay attention to this issue in future research.
Point 4: Do the results tell us anything about whether MIER is superior to CACER in increasing GTFP? Does Table 7 tell us anything about the significance and magnitude of including OFDI as a mediating factor for analyzing the impact of ER on GTFP?
Response 4:Thank you for your suggestion. We've already added the”whether MIER is superior to CACER in increasing GTFP?” Please see Line 918-920. We have completed the significance and magnitude of including OFDI as a mediating factor for analyzing the impact of ER on GTFP. Please see Line 691.
Point 5:p. 4: The first paragraph of section 2.2 could use some work. I believe that on line 160, "GTFP" should be "TFP". More broadly, around line 175, my understanding is that the Porter hypothesis refers to TFP, innovation, and competitiveness only; I don't know of a "strong" version that refers to GTFP. If the authors do -- perhaps the cited Zhang, et al. paper? -- please explain and cite.
Response 5:Yes, you said "TFP" on line 166. Porter hypothesis refers to TFP, innovation and competitiveness by environmental regulation. We have corrected the Porter hypothesis according to your requirements. Please see Line 175.
Point 6:pp. 4-5: In section 2.3, is the ER under examination always in the home country only, or is the paper sometimes referring to ER in the host country? Again, just make clearer, please.
Response 6:The host country in "is an error." We unified all into the home country. Please see Line 219.
Point 7:pp. 14-15: So from 2006-2019, GTFP decreased steadily in all regions? This sounds like a result that should receive more emphasis.
Response 7:The description here is really not very reasonable, We have revised it in the article. Please see Line 550.
Point 8:p. 15: Are provincial leaders rewarded, either formally or informally, for good environmental performance? Might this lead to biased reporting from the province to the center?
Response 8:At present, in the society where information spreads faster and faster, the supervision of the public and the media is more perfect. The reports are based on facts.
Point 9:pp. 16-17: This is a pretty long paper. I would omit Figures 3-6.
Response 9:This is indeed a long article, and We have removed figures 3-6
Point 10:p. 18, line 641: "columns (2)-(6)" should be "columns (4)-(6)".
Response 10:The flaw you raised is existing in my article, We have changed the contents here to "columns (4)-(6)". please see line 629.
Point 11:p. 22: Where, exactly, are the results described in the first sentence shown? Which coefficient or coefficients should I be looking at?
Response 11: The content description here is not very clear, We have added to the content "Columns (3)-(4) and (7)-(8) are the estimated results when OFDI is introduced". Please see line 727-729.
Point 12:p. 26, line 886: I thought the paper was examining OFDI as a mediating influence of ER on TFP. Where did this discussion of CACER as a mediating factor come from? But perhaps this is just an error of discussion.
Response 12:I have read over the modification suggestions you put forward. It should be the mediating effect of CACER. Please see line 880.
Point 13: line 944: I would omit the statement that a policy implication of the paper is that the government should be providing tax incentives to encourage OFDI. If the rationale is to increase GTFP, surely this indirect instrument should be compared with more direct instruments such as tougher ER or incentives for PC research before reaching such a policy conclusion.
Response 13:Your suggestions are very valuable to me. We was not as comprehensive as you considered before. We have revised this part. Omitted the statement that a policy implication of the paper is that the government should be providing tax incentives to encourage OFDI.
Reviewer 2 Report
I enjoyed reading this paper. I have only some minor concerns.
The paper is a bit long but on a fairly mainstream topic. It might be even divided in two papers.
Please, check style and typo errors. See, for instance:
- Line 11: please, check the acronym “CACRER”. It is not the same as in the next lines.
- Line 392: “More details about the decomposition please refer to Banker et al. [71].” Do you mean “For more details about…”
Line 403: what do you mean by “energy structure” and “human capital level”? Please, provide a definition for them when they are introduced fot the first time (now they are described in section 4.2.3.3.). My suggestion is to add a dedicated section in which all the independent variables included in the regression equation are presented.
DEA provides deterministic estimates of efficiency and productivity indicators. Literature suggests that, because of their nature, these indicators cannot be used as dependent variables in a regression equation. Some scholars suggested to adopt a bootstrapping procedure to generate stochastic variables to use in statistical inference. How did authors deal with such a methodological issue? To what extent independent variables do not affect the shape of the production frontier (…and the spaces of efficiency variable and the independent variables are really separated)?
Author Response
Dear reviewer, thank you very much for your valuable comments and suggestions. According to your proposal, we have revised the paper, to get your approval.
Point 1: Line 11: please, check the acronym “CACRER”. It is not the same as in the next lines.
Response 1: Thank you for your suggestion. This is indeed a mistake in writing. We have corrected the article recently and corrected it to "CACER". Please see Line 10.
Point 2: Line 392: “More details about the decomposition please refer to Banker et al. [71].” Do you mean “For more details about…”
Response 2: This is a mistake in my translation. We changed the content to “ The details about the decomposition, i refer to Banker et al.[71]”. Please see Line 405.
Point 3: Line 403: what do you mean by “energy structure” and “human capital level”? Please, provide a definition for them when they are introduced fot the first time (now they are described in section 4.2.3.3.). My suggestion is to add a dedicated section in which all the independent variables included in the regression equation are presented.
Response 3: Your suggestion is very good. We have already added a dedicated section in which all the independent variables included in the regression equation are presented. Please see Line 515-516.
Point 4: DEA provides deterministic estimates of efficiency and productivity indicators. Literature suggests that, because of their nature, these indicators cannot be used as dependent variables in a regression equation. Some scholars suggested to adopt a bootstrapping procedure to generate stochastic variables to use in statistical inference. How did authors deal with such a methodological issue? To what extent independent variables do not affect the shape of the production frontier (…and the spaces of efficiency variable and the independent variables are really separated)?
Response 4:We admit that it is a consensus among scholars that efficiency variables and spatial variables cannot be put together. In order to solve the problem of methodology, We refer to the modeling process of others.
Xie, R. H., Yuan, Y. J., & Huang, J. J. (2017). Different types of environmental regulations and heterogeneous influence on “green” productivity: evidence from China. Ecological economics, 132, 104-112.
Round 2
Reviewer 1 Report
Sorry, but the response to my first (and most important) point is inadequate. The issue is not whether environmental investment reflects ER; the issue is whether it reflects command-and-control (CACER) regulation vis-à-vis market incentive regulation (MIER). One of the sources listed, Xie, et al. (2016), specifically makes this point: “Neither PACE [pollution abatement and control expenditures] nor PAF [pollution abatement fees] enables us to divide formal regulations into two different types mentioned above [‘command-and-control’ vs. market-based].” On the other hand, seemingly paradoxically and without further discussion, those authors then use “Environmental Investments in New Construction Projects (EI) as a proxy for stringency of the command-and-control regulation, and pollutant discharge
fees (PDF) as a proxy for the market-based regulation.” Puzzling.
The Hamamoto (2006) paper cited does not seem relevant to this discussion. It simply uses “pollution control expenditures” as a proxy for “stringency of environmental regulations”.
I remain of the opinion that the right strategy here would be to examine the regulatory regimes of the 30 provinces and categorize them as CACER or MIER (or perhaps, as Xie, et al., suggest, “informal”). The authors say that they like the idea but that it would be a new paper. I’m not so sure; it seems to me potentially a superior (and direct rather than indirect) measure of the primary distinction examined in the paper.
A second-best solution might be the following, and it might require further modeling and estimation (or at least further discussion). As I mentioned in my point 3, the proxy used in the paper for MIER, “total amount of pollutant discharge fees”, may not be a bad one – though I would like to hear more detail. (For example, does it include penalties levied under CACER? That would harm it as a proxy. Does it somehow include the proceeds of emissions right trading systems? If not, that would also harm it as a proxy.) Subject to the details just mentioned, the absence or very small value of these fees might indicate the absence of MIER. Then, If we can assume – can we? – that CACER is the default regime, enforced to some degree by the central authorities, then the level of environmental investment may not be necessary at all – though it may, as in Hamamoto, give some idea of the intensity of the baseline, presumably CACER, regulation (controlling for industry mix, of course).
Regarding my point 7, I would still recommend more discussion of the finding that GTFP has been steadily declining over time. If you believe that result or if you do not, it would still seem to merit more attention, especially since China’s very high (conventionally measured) economic growth over recent years is so widely noted and commented on.
Author Response
Dear reviewer, thank you very much for your valuable comments and suggestions. According to your proposal, we have revised the paper to get your approval.
Point 1:Sorry, but the response to my first (and most important) point is inadequate. The issue is not whether environmental investment reflects ER; the issue is whether it reflects command-and-control (CACER) regulation vis-à-vis market incentive regulation (MIER). One of the sources listed, Xie, et al. (2016), specifically makes this point: “Neither PACE [pollution abatement and control expenditures] nor PAF [pollution abatement fees] enables us to divide formal regulations into two different types mentioned above [‘command-and-control’ vs. market-based].” On the other hand, seemingly paradoxically and without further discussion, those authors then use “Environmental Investments in New Construction Projects (EI) as a proxy for stringency of the command-and-control regulation, and pollutant discharge fees (PDF) as a proxy for the market-based regulation.” Puzzling.
Response 1:Thank you for your valuable advice. We think environmental protection investment reflects command-and-control (CACER) regulation vis-a-vis market incentive regulation (MIER). The term "pollution treatment investment" refers to the funding used to create fixed assets in the process of industrial pollution treatment, including investments in "three simultaneous" environmental protection projects, new and old industrial pollution source control projects, and the construction of urban environmental infrastructure. The term "three simultaneous" refers to a project in which the main component of the project and the infrastructures for preventing pollution are concurrently developed, built, and employed. Most environmental protection investment is derived from government expenditure. We have added this part to the article. Please see Line 506-517.
We also added a reference for the same proxy variable.
- Qiu, Shilei, Zilong Wang, and Shuaishuai Geng. "How do environmental regulation and foreign investment behavior affect green productivity growth in the industrial sector? An empirical test based on Chinese provincial panel data." Journal of Environmental Management 2021, 287: 112282.
However, PAF(pollution abatement and control expenditures) include the costs and expenditures related to abatement operation, maintenance, supervision, tests and inspection, and pollutant emission fees. PACE(pollution abatement fees) gives the sum of control equipment expenditure on wastewater, waste gas, disposal, and noise. Unfortunately, our dataset may not includes detailed information about the expenditures of pollution abatement. So, Xie uses “Environmental Investments in New Construction Projects (EI) as a proxy for the stringency of the command-and-control regulation, and pollutant discharge fees (PDF) as a proxy for the market-based regulation.”
In the article Yang C H, Tseng Y H, Chen C P. Environmental regulations, induced R&D, and productivity: Evidence from Taiwan's manufacturing industries[J]. Resource and Energy Economics, 2012, 34(4): 514-532. The author says, “However, owing to the availability of information, these studies adopt either abatement capital cost or abatement operating cost. Fortunately, our dataset includes detailed information about the expenditures of pollution abatement, which enables us to divide the total PACE into abatement capital and abatement fees. ”
Point 2:I remain of the opinion that the right strategy here would be to examine the regulatory regimes of the 30 provinces and categorize them as CACER or MIER (or perhaps, as Xie, et al., suggest, “informal”). The authors say that they like the idea but that it would be a new paper. I’m not so sure; it seems to me potentially a superior (and direct rather than indirect) measure of the primary distinction examined in the paper.
Response 2:Thank you for your suggestions. The right strategy of examining the regulatory regimes of the 30 provinces and categorizing them as CACER or MIER is a very good idea, and we will apply this method to other articles in future research. We have added this part to the article. Please see Lines 280-313.
Point 3:A second-best solution might be the following, and it might require further modeling and estimation (or at least further discussion). As I mentioned in my point 3, the proxy used in the paper for MIER, “total amount of pollutant discharge fees”, may not be a bad one – though I would like to hear more detail. (For example, does it include penalties levied under CACER? That would harm it as a proxy. Does it somehow include the proceeds of emissions right trading systems? If not, that would also harm it as a proxy.) Subject to the details just mentioned, the absence or very small value of these fees might indicate the absence of MIER. Then, If we can assume – can we? – that CACER is the default regime, enforced to some degree by the central authorities, then the level of environmental investment may not be necessary at all – though it may, as in Hamamoto, give some idea of the intensity of the baseline, presumably CACER, regulation (controlling for industry mix, of course)
Response 3:Thank you for suggesting that "total amount of pollutant discharge fees" is not a bad proxy variable. We have further discussed "total amount of pollutant discharge fees" in the article. The "total amount of pollutant discharge fees" is the compensation for the damage caused by the polluter to the environment. Pollution emission rights reflect the value of the environmental resources occupied. Pollution emission rights trading systems are to control enterprises' emission volume under the premise of total volume control, and enterprises can reduce their emission volume by means of treatment and technological improvement, and their excess emission can be traded as emission rights. We have added the introduction of (MIER) content. Please see Lines 522-537.
Point 4:Regarding my point 7, I would still recommend more discussion of the finding that GTFP has been steadily declining over time. If you believe that result or if you do not, it would still seem to merit more attention, especially since China’s very high (conventionally measured) economic growth over recent years is so widely noted and commented on.
Response 4:Thank you for your suggestions. China's rapid economic growth in recent years has attracted the attention of many scholars. We fully agree with your point of view, and we have added a discussion on the steady decline of GTFP in the article. Please see Lines 602-606.
